# LOOK CAREFULLY: ADAPTIVE VISUAL REINFORCEMENTS IN MULTIMODAL LARGE LANGUAGE MODELS FOR HALLUCINATION MITIGATION

**Xingyu Zhu**[1,2]**, Kesen Zhao**[2]**, Liang Yi**[1]**, Shuo Wang**[1]**, Zhicai Wang**[1]**,**
**Beier Zhu**[1*]**, Hanwang Zhang**[2]**, Xiangnan He**[1*]
[1] MoE Key Lab of BIPC, University of Science and Technology of China
[2] Nanyang Technological University
`xyzhuxyz@mail.ustc.edu.cn, beier.zhu@ustc.edu.cn`

## ABSTRACT

Multimodal large language models (MLLMs) have achieved remarkable progress in vision–language reasoning, yet they remain vulnerable to hallucination, where generated content deviates from the visual evidence. Existing mitigation strategies either demand costly supervision during training or introduce additional latency at inference. Recent vision-enhancement methods attempt to address this by reinforcing visual tokens during decoding, but they typically inject all tokens indiscriminately, leading to interference from background regions and distracting the model from critical cues. To overcome this challenge, we propose an **A**daptive v**I**sual **R**einforcement framework for MLLMs, dubbed as **AIR**. AIR consists of two main components: prototype-based token reduction, which condenses the large pool of visual tokens into a compact subset to suppress redundancy, and OT-guided patch reinforcement, which quantifies the alignment between hidden state and patch embeddings to selectively integrate the most consistent patches into the feed-forward layers. As a result, AIR enhances the model's reliance on salient visual information and effectively mitigates hallucination. Extensive experiments across representative MLLMs demonstrate that AIR substantially reduces hallucination while preserving general capabilities, establishing it as an effective and independent solution for building reliable MLLMs.

## 1 INTRODUCTION

Multimodal large language models (MLLMs) (Chen et al., 2023; Zhu et al., 2024a;c; Liu et al., 2024a; Li et al., 2025; Liu et al., 2024b; Han et al., 2025; Wu et al., 2025e;a;b;d) have achieved remarkable progress by unifying vision and language, enabling reasoning over interleaved text–image inputs. They have been widely applied in tasks (Wu & Yang, 2024; Wu et al., 2025c) such as visual question answering (Wu et al., 2025a) and image captioning (Yang et al., 2023; Zhao et al., 2025). Despite these advances, MLLMs remain prone to hallucination (Jiang et al., 2025; Zheng et al., 2025; Yang et al., 2025a), where generated content is inconsistent with the visual input, *e.g.*, describing non-existent objects or producing contradictory interpretations. This vulnerability poses a barrier to deployment in real-world scenarios.

Existing hallucination mitigation strategies can be broadly divided into training-time and inference-time methods. Training-time approaches (Gunjal et al., 2024; Lyu et al., 2024; Fu et al., 2025; Yang et al., 2025b) rely on additional annotations to fine-tune MLLMs, while inference-time approaches typically adopt contrastive decoding or reranking. Although effective, they either require costly supervision or introduce extra latency. Recent efforts (Fazli et al., 2025; Zheng & Zhang, 2025; Yin et al., 2025; Zou et al., 2024) have strengthened the contribution of image tokens during decoding. They are annotation-free and incur little additional overhead, making them broadly applicable in practice. Concretely, these methods improve grounding by re-injecting visual tokens into the feed-forward network (FFN) (Zou et al., 2024; Yuan et al., 2025; Wan et al., 2025; Zhou et al., 2025).

---

*Corresponding authors.

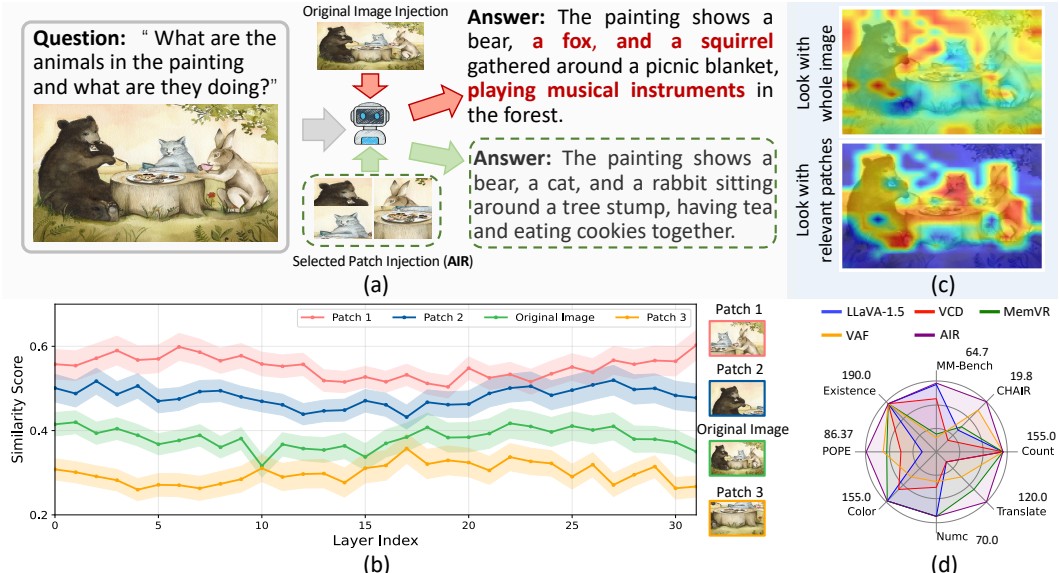

Figure 1: Analysis of existing hallucination mitigation strategies in multimodal large language models (MLLMs). **(a)** existing strategies hallucinate non-existent objects, while AIR produces faithful, image-grounded answers. **(b)** Similarity across decoder layers between hidden states and different visual tokens, showing that salient patches consistently achieve higher alignment than irrelevant ones. **(c)** Attention heatmaps comparing existing re-injection with AIR: prior methods spread attention to irrelevant regions, whereas AIR focuses on semantically critical areas. **(d)** AIR reduces hallucination across benchmarks with minimal impact on general multimodal performance.

Despite their success, we observe that visual inputs often contain substantial interference, such as background regions, which may include redundant objects or distracting semantics. Simply *fusing all visual tokens* into the decoding process can distract the model's attention from critical regions. As illustrated in Fig. 1 (a), we compare two injection strategies, The first strategy, original token injection (Zou et al., 2024), leads to hallucinations as the model attends to irrelevant background regions. In contrast, with relevant token re-injection, the model effectively mitigates hallucinations by focusing on the critical visual content. This is also demonstrated by the heatmap depicted in Fig. 1 (c). To explore this phenomenon in more depth, we analyze the similarity between hidden states and visual tokens across decoding layers using LLaVA-1.5-7B in the MSCOCO dataset (Lin et al., 2014). As illustrated in Fig. 1 (b), effective target regions (*e.g.*, patch 1 and patch 2) consistently yield higher similarity, whereas background regions (patch 3) remain low. The similarity of the original image tokens lies between, suggesting that they dilute the importance of salient cues.

Motivated by these findings, we propose **AIR**, an adaptive visual reinforcement framework that amplifies critical evidence and suppresses redundancy. Instead of re-injecting the full set of visual tokens, AIR is built on two key components. The first, prototype-based token reduction, compresses image tokens into a compact subset, filtering out repetitive background signals and reducing computation. The second, OT-guided patch reinforcement, leverages entropically regularized optimal transport to evaluate alignment between hidden states and patch embeddings, ensuring that only well-aligned regions are integrated into the decoder's FFN. Together, these designs enable the model to focus on salient regions and mitigate hallucination in a training-free and efficient manner.

To validate the effectiveness of our framework, we conduct extensive experiments on multiple representative MLLMs, including LLaVA-1.5-7B (Liu et al., 2024a), Qwen-VL Bai et al. (2023), and GLM-4V-9B Zeng et al. (2024). The results demonstrate that AIR consistently lowers hallucination rates on benchmarks such as CHAIR (Rohrbach et al., 2018) and POPE (Li et al., 2023b), while maintaining strong performance on complementary tasks including existence, counting, and translation, as illustrated in Fig. 1 (d). These results highlight that AIR is a training-free framework that generalizes across diverse MLLMs, providing an effective solution for hallucination mitigation.

## 2 RELATED WORKS

**Hallucination in MLLMs.** Object hallucination occurs when multimodal LLMs generate fluent but visually inconsistent outputs, compromising reliability in multimodal reasoning. Existing mitigation strategies fall into three categories. *Training-based methods* fine-tune models with curated datasets (Gunjal et al., 2024), enforce cross-modal alignment (Fu et al., 2025), or apply preference optimization (Yang et al., 2025b), but these approaches require costly annotations and heavy computation. *Post-processing approaches* revise or filter responses with external models or lightweight revisers, such as LURE (Zhou et al., 2024; Yin et al., 2024; Wu et al., 2024; Yang et al., 2025a; Liu et al., 2025), which increase flexibility but add system complexity. *Inference-time interventions* modify decoding without retraining, for example logit shifting (Zhao et al., 2024) or contrastive decoding (Leng et al., 2024; Wang et al., 2025), providing efficiency but sometimes reducing stability. Different from these paradigms, we introduce a training-free approach that adaptively calibrates attention and selectively reinforces critical visual patches via optimal transport, achieving effective hallucination mitigation without fine-tuning or auxiliary models.

**Optimal transport.** Optimal transport (OT) provides a principled framework to measure discrepancies between distributions by explicitly modeling the cost of transporting probability mass (Monge, 1781). Unlike pointwise similarity metrics (*e.g.*, cosine distance), OT captures the global geometric structure of two distributions, yielding a more faithful measure of semantic alignment. Although the exact solution is computationally demanding, entropic regularization with the Sinkhorn algorithm (Cuturi, 2013) has made OT scalable to high-dimensional problems. These advances have enabled a broad range of applications, including domain adaptation (Turrisi et al., 2022; Chang et al., 2022), distribution calibration (Guo et al., 2022), and image recognition/clustering (Wang et al., 2023; Li et al., 2023a; Zhu et al., 2026). In vision–language modeling, OT has been adopted to align modality distributions in few-shot learning (Zhou et al., 2022; Lazarou et al., 2021; Zhu et al., 2025a;b), refine prompts for cross-modal transfer (Chen et al., 2022), and improve zero-shot generalization (Zhu et al., 2024d; Fang et al., 2025; Zhu et al., 2024b; Zhou et al., 2023). In contrast, our approach employs OT directly at inference: we compute the transport distance between original image tokens and patch embeddings, and use it as a fine-grained criterion to select patches that preserve critical visual semantics. The selected patches are then fused into the decoder, providing a lightweight yet distribution-aware reinforcement of visual evidence.

## 3 METHOD

### 3.1 PRELIMINARIES

**Multimodal large language models.** Multimodal large language models (MLLMs) extend conventional large language models (LLMs) to jointly process text and images. An MLLM typically consists of a vision encoder, a text encoder, and an autoregressive decoder. Given a textual query $x = [x_1, \ldots, x_L]$ and an image $v$, the vision encoder extracts visual features and then transforms them into aligned visual tokens $\mathbf{Z} = [\mathbf{z}_1, \mathbf{z}_2, \cdots, \mathbf{z}_K]$. Let $[\mathbf{x}_1, \cdots, \mathbf{x}_L]$ denote the embeddings of $x$ produced by the text encoder, and define the multimodal input sequence:

$$\mathbf{X} = [\mathbf{Z}, \mathbf{x}_1, \ldots, \mathbf{x}_L]. \tag{1}$$

At decoding step $t$, the transformer-based decoder produces unnormalized logits $f_\theta(\cdot \mid \mathbf{X}, y_{<t})$, from which the next-token distribution is obtained via the softmax:

$$p_\theta(\cdot \mid \mathbf{X}, y_{<t}) = \text{softmax}(f_\theta(\cdot \mid \mathbf{X}, y_{<t})), \qquad y_t \sim p_\theta(\cdot \mid \mathbf{X}, y_{<t}), \ \ t = 1, \ldots, T. \tag{2}$$

Here, $\theta$ denotes all model parameters and $y_{<t}$ is the previously generated tokens.

**Optimal transport.** Optimal Transport (OT) (Monge, 1781; Wang et al., 2023) offers a principled framework for quantifying the discrepancy between two probability distributions. Consider two discrete measures in the feature space: $\mathbb{P} = \sum_{i=1}^{|V|} a_i \delta(\mathbf{v}_i - \mathbf{v})$ and $\mathbb{Q} = \sum_{j=1}^{|U|} b_j \delta(\mathbf{u}_j - \mathbf{u})$, where $\delta$ denotes the Dirac delta function, and $|V|$ and $|U|$ are the number of support points in $\mathbb{P}$ and $\mathbb{Q}$, respectively. Here, $\mathbf{a} = [a_1, \ldots, a_{|V|}]^\top$ and $\mathbf{b} = [b_1, \ldots, b_{|U|}]^\top$ are probability vectors that sum to one. Given a cost matrix $\mathbf{C} \in \mathbb{R}^{|V| \times |U|}$, where $\mathbf{C}(i, j)$ is the element in $\mathbf{C}$, denoting the cost of transporting unit mass from $\mathbf{v}_i$ to $\mathbf{u}_j$, and the OT distance between $\mathbb{P}$ and $\mathbb{Q}$ is formulated as:

$$d_{\text{OT}}(\mathbb{P}, \mathbb{Q}; \mathbf{C}) = \min_{\mathbf{T}} \langle \mathbf{T}, \mathbf{C} \rangle, \quad \text{s.t. } \mathbf{T}\mathbf{1}_{|U|} = \mathbf{a}, \ \ \mathbf{T}^\top \mathbf{1}_{|V|} = \mathbf{b}, \tag{3}$$

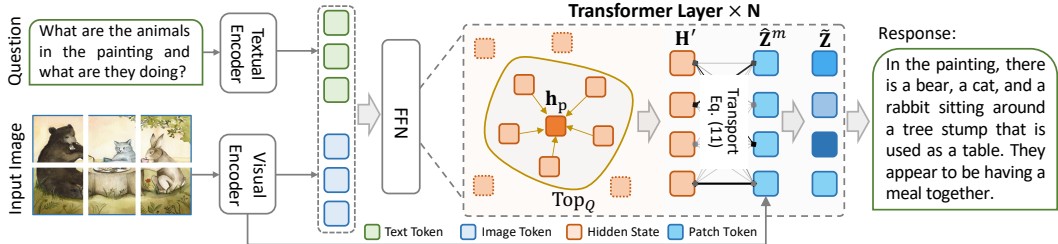

Figure 2: Overview of our proposed AIR framework. Given a multimodal input (image and question), visual features are extracted by the visual encoder and aligned with textual embeddings via the projector. Inside the Transformer layers, visual tokens are first compressed through prototype-based selection to remove redundancy, and then reinforced by patch-level alignment using optimal transport. These selective reinforcement strategies enrich the hidden states with salient visual cues, enabling AIR to produce safer, more faithful, and visually grounded responses.

where $\mathbf{T} \in \mathbb{R}^{|V| \times |U|}$ is the transport plan specifying how mass is moved between the two distributions. The notation $\langle \cdot, \cdot \rangle$ denotes the Frobenius inner product, and $\mathbf{1}_{|V|}$ is an all-ones vector of dimension $|V|$. Directly solving Eq. (3) is computationally intensive. Prior work (Lazarou et al., 2021; Chen et al., 2022) addresses this by employing the Sinkhorn algorithm (Cuturi, 2013), which introduces entropic regularization to achieve efficient optimization:

$$d_{\text{OT}}(\mathbb{P}, \mathbb{Q}; \mathbf{C}) = \min_{\mathbf{T}} \langle \mathbf{T}, \mathbf{C} \rangle - \epsilon h(\mathbf{T}), \tag{4}$$

where $h(\cdot)$ denotes the entropy and $\epsilon \geq 0$ controls the strength of the regularization.

## 3.2 ADAPTIVE VISUAL REINFORCEMENT

**Prototype-based token reduction.** Our visual reinforcement strategy operates in the feed-forward network (FFN) of each Transformer block, which consists of two fully connected layers with a non-linear activation:

$$\text{FFN}(\mathbf{H}) = \phi(\mathbf{H}\,\mathbf{W}_1)\,\mathbf{W}_2^{\top}, \tag{5}$$

where $\phi(\cdot)$ denotes the activation function and $\mathbf{H}$ represents the hidden states. As the Transformer goes deeper, attention tends to become progressively biased toward textual tokens, which diminishes the contribution of visual tokens in later layers. A common remedy is to re-inject visual tokens into the FFN (Zou et al., 2024):

$$\text{FFN}(\mathbf{H}|\mathbf{Z}) = \phi(\mathbf{H}\mathbf{Z}^{\top})\mathbf{Z} \tag{6}$$

where $\mathbf{Z}$ denotes the visual tokens aligned by the projector. However, since visual tokens are derived from the entire image, their length $K$ is typically large (e.g., $K = 576$ in LLaVA), which introduces redundancy and noise. Directly re-injecting all tokens not only incurs unnecessary computational overhead but also prevents the model from focusing on the most informative regions. To address this issue, we first condense $\mathbf{H}$ into a compact subset. We compute a prototype $\mathbf{h}_{\text{p}} = \frac{1}{K}\sum_{k=1}^{K} \mathbf{h}_k$ as a coarse summary of the visual semantics, and rank tokens by their distance to this prototype:

$$d(\mathbf{h}_k, \mathbf{h}_{\text{p}}) = \|\mathbf{h}_k - \mathbf{h}_{\text{p}}\|_2, \qquad k = 1, \ldots, K. \tag{7}$$

Tokens with larger distances encode more distinctive cues not captured by the global prototype. We therefore retain only the $\text{Top}_Q$ tokens:

$$\mathbf{H}' \leftarrow \{\mathbf{h}_k \mid k \in \text{Top}_Q(\{d(\mathbf{h}_k, \mathbf{h}_{\text{p}})\})\}, \tag{8}$$

ensuring that subsequent reinforcement operates on a compact set of visual representations.

**OT-guided patch reinforcement.** While prototype selection reduces global redundancy, different image regions may still vary in importance. To further emphasize critical details, we crop the image into multiple patches $\{\hat{v}^m\}_{m=1}^{M}$ with corresponding embeddings $\{\hat{\mathbf{Z}}^m\}_{m=1}^{M}$, where each $\hat{\mathbf{Z}}^m = [\mathbf{z}_1^m, \mathbf{z}_2^m, \cdots, \mathbf{z}_N^m]$ encodes fine-grained visual details. We model the original hidden states and patch-level tokens as discrete distributions:

$$\mathbb{P}(\mathbf{h}) = \sum_{k=1}^{Q} a_k \delta(\mathbf{h}_k - \mathbf{h}), \quad \mathbb{Q}_m(\hat{\mathbf{z}}) = \sum_{n=1}^{N} b_n^m \delta(\hat{\mathbf{z}}_n^m - \hat{\mathbf{z}}), \tag{9}$$

where $\delta(\cdot)$ denotes the Dirac delta function, and $a_k$, $b_n^m$ are normalized importance weights. The alignment between $\mathbb{P}$ and $\mathbb{Q}_m$ is quantified using the OT distance:

$$d_{\text{OT}}(\mathbb{P}, \mathbb{Q}_m; \mathbf{C}_m) = \min_{\mathbf{T}_m \geq 0} \langle \mathbf{T}_m, \mathbf{C}_m \rangle, \quad \text{s.t.} \quad \mathbf{T}_m \mathbf{1}_Q = \mathbf{a}, \ \mathbf{T}_m^\top \mathbf{1}_N = \mathbf{b}_m, \qquad (10)$$

where $\mathbf{a} = [\frac{1}{Q}, \ldots, \frac{1}{Q}]^\top$, $\mathbf{b}_m = [\frac{1}{N}, \ldots, \frac{1}{N}]^\top$, and $\mathbf{C}_m(k, n) = 1 - \cos(\mathbf{z}_k, \hat{\mathbf{z}}_n^m)$. The transport plan $\mathbf{T}_m$ is efficiently obtained via the Sinkhorn-Knopp algorithm (Zhu et al., 2024d; Chen et al., 2022). For each patch $m$, we compute an aggregated OT distance:

$$d_{\text{OT}}(m) = \sum_{k=1}^{Q} \sum_{n=1}^{N} \mathbf{T}_m(k, n) \, \mathbf{C}_m(k, n). \qquad (11)$$

A lower OT distance indicates stronger alignment with the original image and thus suggests that the patch retains more critical visual information. We therefore select patches with threshodling $\tau$:

$$\mathcal{M} = \{ m \mid d_{\text{OT}}(m) \leq \tau \}. \qquad (12)$$

The embeddings of these selected patches are then fused with the original image tokens:

$$\tilde{\mathbf{Z}} \leftarrow \text{concat}\big(\{\hat{\mathbf{Z}}^m \mid m \in \mathcal{M}\}\big). \qquad (13)$$

This selective fusion enhances the representation with critical visual details, strengthening the role of image information in the hidden states. The final re-injection into the FFN is formulated as:

$$\text{FFN}(\mathbf{H}|\tilde{\mathbf{Z}}) = \phi(\mathbf{H} \, \mathbf{W}_1) \, \mathbf{W}_2^\top + \phi(\mathbf{H}' \tilde{\mathbf{Z}}^\top)\tilde{\mathbf{Z}} \qquad (14)$$

### 3.3 THEORETICAL ANALYSIS

To validate the effectiveness of our OT-based patch selection in Dynamic Token Infusion, we compare the sensitivity of our OT distance metric $d_{\text{OT}}(m)$ with the baseline cosine distance $d_{\cos}(m) = \frac{1}{KN} \sum_{k=1}^{K} \sum_{n=1}^{N} \mathbf{C}_m(k, n)$. In our framework, a patch $m$ is selected as useful if $d_{\text{OT}}(m) \leq \tau$, as lower OT distances indicate stronger alignment with critical visual semantics. Conversely, for the cosine baseline, a patch is selected if $d_{\cos}(m) \leq \tau_{\cos}$, reflecting misalignment. We prove that the OT-based method achieves strictly higher sensitivity in distinguishing patches:

$$|d_{\text{OT}}(m_1) - d_{\text{OT}}(m_2)| > |d_{\cos}(m_1) - d_{\cos}(m_2)|, \qquad (15)$$

except in the degenerate case where $\mathbf{C}_{m_1} = \mathbf{C}_{m_2}$.

**Why OT enhances patch selection sensitivity.** The OT-based metric employs an adaptive transport plan $\mathbf{T}_m$, computed via Sinkhorn-Knopp, prioritizing low-cost alignments (high cosine similarities) between original and patch tokens. This adaptive weighting amplifies differences in semantic alignment, resulting in a larger separation $|d_{\text{OT}}(m_1) - d_{\text{OT}}(m_2)|$ compared to the uniform weighting of $d_{\cos}$, which averages all pairwise costs and dilutes discriminative features. The increased sensitivity ensures more precise identification of patches in $\mathcal{M}$ that capture critical visual information. A proof is given in Appendix C.

## 4 EXPERIMENTS

In this section, we present the experimental results of our method across hallucination and general benchmarks, including performance comparisons, ablation studies, and visualization analyses.

### 4.1 EXPERIMENTAL SETUP

**Base model and baselines.** We validate our method on three MLLMs, including LLaVA-1.5-7B (Liu et al., 2024a), Qwen-VL-Chat (Bai et al., 2023), and GLM-4V-9B (Zeng et al., 2024). For comparison, we include several state-of-the-art object hallucination mitigation methods: VCD (Leng et al., 2024), MemVR (Zou et al., 2024), and VAF (Yin et al., 2025).

Table 1: CHAIR evaluation results on MSCOCO dataset of MLLMs with different methods. We use 64 as the maximum token in this experiment. Bold indicates the best result of all methods.

| Method | LLaVA-1.5-7B | | | Qwen-VL-Chat | | | GLM-4V-9B | | |
|---|---|---|---|---|---|---|---|---|---|
| | CHAIR$_S$ ↓ | CHAIR$_I$ ↓ | BLEU↑ | CHAIR$_S$ ↓ | CHAIR$_I$ ↓ | BLEU↑ | CHAIR$_S$ ↓ | CHAIR$_I$ ↓ | BLEU↑ |
| Vanilla | 22.0 $_{\uparrow 0.0}$ | 6.7 $_{\uparrow 0.0}$ | 14.5 $_{\uparrow 0.0}$ | 20.0 $_{\uparrow 0.0}$ | 6.2 $_{\uparrow 0.0}$ | 13.5 $_{\uparrow 0.0}$ | 13.0 $_{\uparrow 0.0}$ | 5.6 $_{\uparrow 0.0}$ | **9.8** $_{\uparrow 0.0}$ |
| VCD | 24.6 $_{\uparrow 2.6}$ | 7.3 $_{\uparrow 0.6}$ | 13.9 $_{\downarrow 0.6}$ | 19.2 $_{\downarrow 0.8}$ | **5.7** $_{\downarrow 0.5}$ | 13.4 $_{\downarrow 0.1}$ | 14.8 $_{\uparrow 1.8}$ | 6.5 $_{\uparrow 0.9}$ | 9.5 $_{\downarrow 0.3}$ |
| MemVR | 21.6 $_{\downarrow 0.4}$ | 6.4 $_{\downarrow 0.3}$ | 14.4 $_{\downarrow 0.1}$ | 20.0 $_{\uparrow 0.0}$ | 6.1 $_{\downarrow 0.1}$ | 13.3 $_{\downarrow 0.2}$ | 13.0 $_{\uparrow 0.0}$ | 5.6 $_{\uparrow 0.0}$ | **9.8** $_{\uparrow 0.0}$ |
| VAF | 20.4 $_{\downarrow 1.6}$ | 6.5 $_{\downarrow 0.2}$ | **14.6** $_{\uparrow 0.1}$ | 20.6 $_{\uparrow 0.6}$ | 6.6 $_{\uparrow 0.4}$ | 13.4 $_{\downarrow 0.1}$ | **11.6** $_{\downarrow 1.4}$ | **5.3** $_{\downarrow 0.3}$ | 9.7 $_{\downarrow 0.1}$ |
| **AIR** | **18.4** $_{\downarrow 3.6}$ | **5.7** $_{\downarrow 1.0}$ | 14.4 $_{\downarrow 0.1}$ | **18.6** $_{\downarrow 1.4}$ | 5.9 $_{\downarrow 0.3}$ | **13.6** $_{\uparrow 0.1}$ | **11.6** $_{\downarrow 1.4}$ | **5.3** $_{\downarrow 0.3}$ | 9.7 $_{\downarrow 0.1}$ |

Table 2: Performance on POPE benchmark using LLaVA-1.5-7B. Bold numbers indicate the best results. We report accuracy and F1-score under three settings, *i.e.*, *Random*, *Popular*, and *Adversarial*, to show the robustness of different methods.

| Datasets | Methods | Random | | Popular | | Adversarial | |
|---|---|---|---|---|---|---|---|
| | | Accuracy ↑ | F1-score ↑ | Accuracy ↑ | F1-score ↑ | Accuracy ↑ | F1-score ↑ |
| MSCOCO | Vanilla | 83.7 $_{\uparrow 0.0}$ | 83.0 $_{\uparrow 0.0}$ | 78.2 $_{\uparrow 0.0}$ | 78.4 $_{\uparrow 0.0}$ | 75.0 $_{\uparrow 0.0}$ | 76.0 $_{\uparrow 0.0}$ |
| | VCD | 85.4 $_{\uparrow 1.7}$ | 83.7 $_{\uparrow 0.7}$ | 84.3 $_{\uparrow 6.1}$ | 83.0 $_{\uparrow 4.6}$ | 81.8 $_{\uparrow 6.8}$ | 80.9 $_{\uparrow 4.9}$ |
| | MemVR | 87.6 $_{\uparrow 3.9}$ | 86.2 $_{\uparrow 3.2}$ | 86.0 $_{\uparrow 7.8}$ | 84.7 $_{\uparrow 6.3}$ | 83.5 $_{\uparrow 8.5}$ | 82.5 $_{\uparrow 6.5}$ |
| | VAF | 87.6 $_{\uparrow 3.9}$ | 86.2 $_{\uparrow 3.2}$ | 86.2 $_{\uparrow 8.0}$ | 85.0 $_{\uparrow 6.6}$ | **83.9** $_{\uparrow 8.9}$ | 82.8 $_{\uparrow 6.8}$ |
| | **AIR** | **89.0** $_{\uparrow 5.3}$ | **88.2** $_{\uparrow 5.2}$ | **87.1** $_{\uparrow 8.9}$ | **86.4** $_{\uparrow 8.0}$ | **83.9** $_{\uparrow 8.9}$ | **83.6** $_{\uparrow 7.6}$ |
| A-OKVQA | Vanilla | 83.4 $_{\uparrow 0.0}$ | 82.6 $_{\uparrow 0.0}$ | 79.9 $_{\uparrow 0.0}$ | 79.6 $_{\uparrow 0.0}$ | 74.0 $_{\uparrow 0.0}$ | 75.1 $_{\uparrow 0.0}$ |
| | VCD | 85.9 $_{\uparrow 2.5}$ | 85.4 $_{\uparrow 2.8}$ | 81.9 $_{\uparrow 2.0}$ | 82.0 $_{\uparrow 2.4}$ | 76.7 $_{\uparrow 2.7}$ | 78.4 $_{\uparrow 3.3}$ |
| | MemVR | 89.0 $_{\uparrow 5.6}$ | 88.5 $_{\uparrow 5.9}$ | **84.6** $_{\uparrow 4.8}$ | 84.6 $_{\uparrow 5.1}$ | **78.3** $_{\uparrow 4.3}$ | 79.6 $_{\uparrow 4.5}$ |
| | VAF | 88.7 $_{\uparrow 5.3}$ | 88.3 $_{\uparrow 5.7}$ | 84.1 $_{\uparrow 4.2}$ | 84.3 $_{\uparrow 4.7}$ | 76.9 $_{\uparrow 2.9}$ | 78.7 $_{\uparrow 3.6}$ |
| | **AIR** | **89.2** $_{\uparrow 5.8}$ | **88.9** $_{\uparrow 6.3}$ | 84.4 $_{\uparrow 4.5}$ | **84.7** $_{\uparrow 5.1}$ | 78.0 $_{\uparrow 4.0}$ | **79.7** $_{\uparrow 4.6}$ |
| GQA | Vanilla | 83.7 $_{\uparrow 0.0}$ | 83.0 $_{\uparrow 0.0}$ | 78.2 $_{\uparrow 0.0}$ | 78.4 $_{\uparrow 0.0}$ | 75.1 $_{\uparrow 0.0}$ | 76.1 $_{\uparrow 0.0}$ |
| | VCD | 86.3 $_{\uparrow 2.6}$ | 85.8 $_{\uparrow 2.8}$ | 78.4 $_{\uparrow 0.2}$ | 79.0 $_{\uparrow 0.6}$ | 76.2 $_{\uparrow 1.1}$ | 77.4 $_{\uparrow 1.3}$ |
| | MemVR | 89.3 $_{\uparrow 5.6}$ | 88.9 $_{\uparrow 5.9}$ | 82.9 $_{\uparrow 4.7}$ | 83.4 $_{\uparrow 5.0}$ | 80.3 $_{\uparrow 5.2}$ | 81.4 $_{\uparrow 5.3}$ |
| | VAF | 88.1 $_{\uparrow 4.4}$ | 87.7 $_{\uparrow 4.7}$ | 79.4 $_{\uparrow 1.2}$ | 80.6 $_{\uparrow 2.2}$ | 78.2 $_{\uparrow 3.1}$ | 79.7 $_{\uparrow 3.6}$ |
| | **AIR** | **89.7** $_{\uparrow 6.0}$ | **89.5** $_{\uparrow 6.5}$ | **83.0** $_{\uparrow 4.8}$ | **83.8** $_{\uparrow 5.4}$ | **80.4** $_{\uparrow 5.3}$ | **81.7** $_{\uparrow 5.6}$ |

**Evaluation benchmarks and metrics.** We evaluate our method on a variety of benchmarks. For hallucination assessment, we use CHAIR (Rohrbach et al., 2018) on 500 randomly sampled MSCOCO (Lin et al., 2014) images, and POPE (Li et al., 2023b) on MSCOCO, A-OKVQA (Schwenk et al., 2022), and GQA (Hudson & Manning, 2019). For general-purpose evaluation, we employ LLaVA-Bench (Liu et al., 2023), MME (Fu et al., 2023), and MMBench (Liu et al., 2024c). As evaluation metrics, we report CHAIR$_S$, CHAIR$_I$, BLEU, accuracy, and F1-score. Detailed descriptions and implementations are provided in the Appendix B.1 and B.2.

## 4.2 PERFORMANCE ON HALLUCINATION BENCHMARKS

As shown in Table 1, AIR consistently achieves the lowest CHAIR$_S$ and CHAIR$_I$ across three representative MLLMs, demonstrating its effectiveness in suppressing hallucinations. For example, on LLaVA-1.5-7B, AIR reduces CHAIR$_S$ from 22.0 to 18.4 and CHAIR$_I$ from 6.7 to 5.7, while maintaining comparable BLEU scores. This confirms that selectively reinforcing salient tokens mitigates hallucinations more reliably than indiscriminate re-injection. Table 2 further validates the robustness of AIR on the POPE benchmark. Across MSCOCO, A-OKVQA, and GQA, AIR achieves the best or near-best accuracy and F1-score under Random, Popular, and Adversarial settings. Notably, AIR sustains strong performance even under adversarial prompts, outperforming prior defenses such as MemVR. More experimental results are reported in Appendix B.3, Tables 8 and 9.

## 4.3 PERFORMANCE ON GENERAL-PURPOSE BENCHMARKS

As shown in Table 3, AIR preserves strong performance on MME and MMBench, achieving results comparable to or better than existing methods across object- and attribute-level tasks. This indicates

Table 3: Comparison of evaluation results on the MME Hallucination subset and MMBench.

| | Methods | MME-Hall | Object-Level | | Attribute-Level | | Cognition | MMBench |
|---|---|---|---|---|---|---|---|---|
| | | Total ↑ | Existence ↑ | Count ↑ | Position ↑ | Color ↑ | Score ↑ | Accuracy ↑ |
| **LLaVA-1.5** | Vanilla | 643.3 ↑0.0 | 190.0 ↑0.0 | 155.0 ↑0.0 | 128.3 ↑0.0 | 170.0 ↑0.0 | 357.8 ↑0.0 | 64.6 ↑0.0 |
| | VCD | 613.3 ↓30. | 190.0 ↑0.0 | 140.0 ↓15. | 118.3 ↓10. | 165.0 ↓5.0 | 337.1 ↓20. | 61.6 ↓3.0 |
| | MemVR | 648.3 ↑5.0 | 190.0 ↑0.0 | 155.0 ↑0.0 | 133.3 ↑5.0 | 170.0 ↑0.0 | 378.6 ↑20. | 64.6 ↑0.0 |
| | VAF | 603.3 ↓40. | 190.0 ↑0.0 | 135.0 ↓20. | 108.3 ↓20. | 170.0 ↑0.0 | 322.8 ↓35. | 14.8 ↓49. |
| | **AIR** | 638.3 ↓5.0 | 190.0 ↑0.0 | 155.0 ↑0.0 | 123.3 ↓5.0 | 170.0 ↑0.0 | 372.5 ↑14. | **64.7** ↑0.1 |
| **Qwen-VL** | Vanilla | 631.7 ↑0.0 | 185.0 ↑0.0 | 145.0 ↑0.0 | 126.7 ↑0.0 | 175.0 ↑0.0 | 342.8 ↑0.0 | 59.9 ↑0.0 |
| | VCD | 626.7 ↓5.0 | 180.0 ↓5.0 | 145.0 ↑0.0 | 126.7 ↑0.0 | 175.0 ↑0.0 | 348.9 ↑6.1 | **60.3** ↑0.4 |
| | MemVR | **636.7** ↑5.0 | 185.0 ↑0.0 | 145.0 ↑0.0 | **131.7** ↑5.0 | 175.0 ↑0.0 | 337.5 ↓5.3 | 60.0 ↑0.1 |
| | VAF | 631.7 ↑0.0 | 185.0 ↑0.0 | 145.0 ↑0.0 | 126.7 ↑0.0 | 175.0 ↑0.0 | 329.3 ↓14. | 60.0 ↑0.1 |
| | **AIR** | **636.7** ↑5.0 | 185.0 ↑0.0 | 145.0 ↑0.0 | 126.7 ↑0.0 | **180.0** ↑5.0 | **352.1** ↑9.3 | 60.0 ↑0.1 |
| **GLM-4V-9B** | Vanilla | **703.3** ↑0.0 | **200.0** ↑0.0 | 168.3 ↑0.0 | **156.7** ↑0.0 | **178.3** ↑0.0 | 479.6 ↑0.0 | **81.3** ↑0.0 |
| | VCD | 696.7 ↓6.6 | **200.0** ↑0.0 | 170.0 ↑1.7 | 151.7 ↓5.0 | 175.0 ↓3.3 | **485.0** ↑5.4 | 80.2 ↓1.1 |
| | MemVR | **703.3** ↑0.0 | **200.0** ↑0.0 | 168.3 ↑0.0 | **156.7** ↑0.0 | **178.3** ↑0.0 | 479.6 ↑0.0 | **81.3** ↑0.1 |
| | VAF | **703.3** ↑0.0 | **200.0** ↑0.0 | 173.3 ↑5.0 | 151.7 ↓5.0 | **178.3** ↑0.0 | 479.6 ↑0.0 | **81.3** ↓5.7 |
| | **AIR** | **703.3** ↑0.0 | **200.0** ↑0.0 | 173.3 ↑5.0 | 151.7 ↓5.0 | **178.3** ↑0.0 | 479.6 ↑0.0 | **81.3** ↑0.1 |

Table 4: Results of GPT-4V-aided evaluation on LLaVA-Bench following the setting in (Leng et al., 2024). Both metrics are on a scale of 10.

| Model | Method | Accuracy↑ | Detailedness↑ |
|---|---|---|---|
| LLaVA-1.5 | Vanilla | 5.59 | 4.72 |
| | **AIR** | **5.83** | **5.12** |
| Qwen-VL-Chat | Vanilla | 5.85 | 4.98 |
| | **AIR** | **6.18** | **5.12** |
| GLM-4V-9B | Vanilla | 6.76 | 5.32 |
| | **AIR** | **6.93** | **5.52** |

Table 5: Performance comparison of LLaVA-1.5-7B when operating different decoding layer ranges with our method.

| $\{\ell\}$ | CHAIR$_S$ ↓ | CHAIR$_I$ ↓ | BLEU↑ |
|---|---|---|---|
| 16-32 | 19.6 | 6.0 | 14.5 |
| 18-32 | 19.2 | 6.0 | 14.4 |
| 20-32 | 19.0 | **5.5** | 14.4 |
| 22-32 | 19.4 | 5.8 | 14.5 |
| 24-32 | 18.8 | 6.0 | 14.4 |
| 26-32 | **18.4** | 5.7 | 14.4 |
| 28-32 | 19.4 | 5.7 | 14.5 |
| 30-32 | 22.0 | 6.7 | 14.5 |

that selective reinforcement does not compromise general reasoning ability. Table 4 further shows that AIR consistently improves GPT-4V-aided evaluation on LLaVA-Bench. Across all models, both Accuracy and Detailedness scores increase, confirming that AIR enhances output quality while maintaining broad multimodal capability. The detailed results on MME and MMBench are provided in Appendix B.3, Table 10 and 11.

## 4.4 ABLATION STUDIES

**Impact of operating layers.** To evaluate how the choice of operating layers influences performance, we conduct an ablation study on the CHAIR dataset with LLaVA-1.5-7B, as reported in Table 5. Following prior work (Zou et al., 2024; Yin et al., 2025; Yang et al., 2025a) that emphasizes the role of mid-to-deep layers in vision–language fusion, we begin our analysis from layer 16 onward. The results indicate that reinforcing visual tokens in the range 24–32 yields the most favorable trade-off, achieving the lowest CHAIR$_S$ (18.8) while keeping CHAIR$_I$ (6.0) and BLEU (14.4) stable. In comparison, applying reinforcement too late (*e.g.*, 30–32) or across overly broad spans (*e.g.*, 18–32) leads to higher hallucination scores, suggesting that very late layers are overly text-biased and wide ranges dilute the effect. These findings confirm that mid-to-deep layers are the most effective operating region for enhancing visual grounding while preserving generation quality.

**Effectiveness of different components.** Table 6 presents the ablation results on CHAIR. Removing both components yields the highest hallucination rates, highlighting the importance of visual reinforcement. Incorporating only prototype-based token reduction reduces CHAIR$_s$ from 22.7 to 22.3, showing that condensing redundant tokens provides modest gains. OT-based patch reinforcement

Table 6: Ablation study of our method on the CHAIR using LLaVA-1.5-7B, where we ablate two alignment components: prototype-based token reduction and OT-based patch reinforcement.

| Model | Prototype-based Token Reduction | OT-based Patch Reinforcement | CHAIR | |
|---|---|---|---|---|
| | | | CHAIR$_S$ ↓ | CHAIR$_I$ ↓ |
| LLaVA-1.5-7B | ✗ | ✗ | 22.7 | 6.7 |
| | ✗ | ✓ | 22.3 | 6.5 |
| | ✓ | ✗ | 20.2 | 6.2 |
| | ✓ | ✓ | **19.8** | **5.8** |

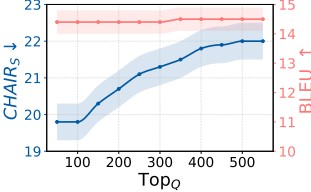 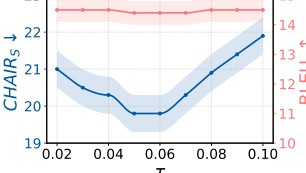 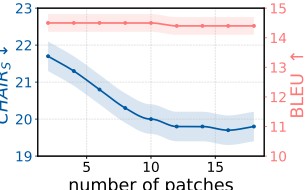

Figure 3: Performance under different numbers of retained visual tokens Top$_Q$.

Figure 4: Performance with varying OT-based distance threshold $\tau$.

Figure 5: Performance as the number of selected image patches increases.

alone achieves a larger improvement (22.7 to 20.2), confirming the effectiveness of distribution-aware patch selection. When both modules are combined, CHAIR$_S$ further decreases to 19.8 and CHAIR$_I$ drops to 5.8, demonstrating that the two components are complementary and jointly contribute to mitigating hallucinations.

**Impact of Top$_Q$ selection.** As shown in Fig. 3, increasing the number of retained visual tokens Top$_Q$ leads to a steady reduction in hallucinations, evidenced by the decline of CHAIR$_s$. This result highlights that prototype-based selection successfully preserves the most discriminative cues while filtering redundancy. Moreover, BLEU scores remain nearly unchanged, indicating that semantic fidelity is maintained without compromising language generation quality.

**Impact of threshold $\tau$.** Fig. 4 illustrates that varying the OT threshold $\tau$ produces a U-shaped performance trend. An overly strict threshold removes informative patches, which weakens visual grounding and increases hallucinations, while an overly loose threshold admits irrelevant regions and introduces noise. A moderate value of $\tau$ achieves the best balance between selectivity and coverage, resulting in reduced CHAIR$_s$ and stable BLEU, thereby confirming the effectiveness of OT-based patch selection.

**Impact of augmented patches.** Fig. 5 quantifies the effect of enlarging the patch pool for OT-based selection: with more candidate patches, the transport plan can match lower-cost, better-aligned regions, yielding stronger visual evidence. Correspondingly, CHAIR$_s$ decreases steadily, while BLEU remains essentially unchanged, indicating that increased candidate coverage improves selected visual information without harming fluency.

## 4.5 ANALYSIS OF OT-BASED PATCH REINFORCEMENT

We evaluate OT-based patch reinforcement on the CHAIR benchmark using LLaVA as the backbone model. Fig. 6 provides both quantitative and qualitative evidence of its superiority over cosine-based selection. In (a), the distribution of margin differentials shows that OT consistently produces larger gaps between safe and unsafe patches, confirming its stronger discriminative ability. The scatter plot in (b) further supports this observation, as the majority of points lie above the $y = x$ line, indicating that OT achieves greater separation across patch-level comparisons. Importantly, Qualitative examples in (c), drawn from LLaVA-Bench, further illustrate that OT-selected patches concentrate on visually salient regions aligned with the image semantics. Together, these results validate that the adaptive transport plan in OT accentuates meaningful cues and reduces redundancy, thereby improving the model's ability to mitigate hallucinations.

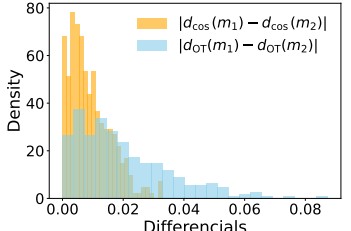
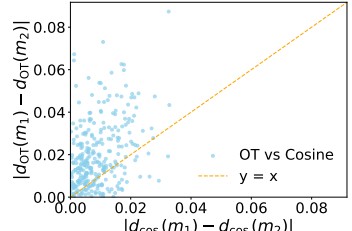
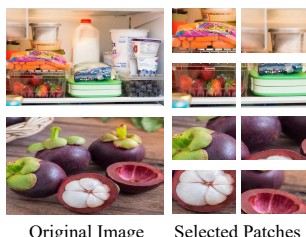

Original Image     Selected Patches

(a) Distribution of margin differentials for OT and cosine distances.
(b) Comparison of margins between OT and cosine distance.
(c) Examples of original images and OT-selected patches.

Figure 6: Analysis of OT-based versus cosine-based patch selection. (a) OT produces consistently larger margin differentials than cosine. (b) Most patch-level results lie above the $y = x$ line, showing that OT provides clearer separation between patches. (c) Qualitative evidence shows that OT focuses on visually informative regions aligned with the original image.

Table 7: Comparison of inference speed, GPU memory usage, and hallucination performance on the CHAIR using a single A100 GPU.

| Model | Avg. Latency ↓ | GPU Memory ↓ | $\text{CHAIR}_S$ ↓ | $\text{CHAIR}_i$ ↓ |
|---|---|---|---|---|
| LLaVA-1.5-7B | 1.68s | 13.5G | 22.0 | 6.7 |
| VAF | 1.69s | 13.5G | 20.4 | 6.5 |
| MemVR | 2.05s | 13.6G | 21.6 | 6.4 |
| **AIR** | 2.07s | 13.7G | 18.4 | 5.7 |

## 4.6 COMPARISON OF EFFICIENCY

The comparison in Table 7 indicates that our method substantially improves safety while preserving efficiency. LLaVA-1.5-7B yields the highest hallucination rates ($\text{CHAIR}_s$=22.0, $\text{CHAIR}_i$=6.7). VAF and MemVR achieve moderate reductions, lowering $\text{CHAIR}_s$ to 20.4 and 21.6, and $\text{CHAIR}_i$ to 6.5 and 6.4, respectively, but with only limited improvements relative to the baseline. In contrast, our framework achieves the lowest hallucination rates ($\text{CHAIR}_s$=18.4, $\text{CHAIR}_i$=5.7), representing a clear gain in both sentence-level and object-level accuracy. These safety improvements come with a slight increase in latency (2.07s vs. 1.68s for LLaVA) and GPU memory (13.7G vs. 13.5G), but the overhead remains marginal compared to the robustness benefits. Overall, the results demonstrate the effectiveness of our approach in suppressing hallucinations while maintaining efficiency.

## 5 LIMITATIONS & FUTURE DISCUSSION

Although AIR demonstrates clear effectiveness in mitigating hallucinations, its application to reasoning multimodal models and agents has not yet been explored. Future work can extend AIR to these broader settings, where adaptive reinforcement may further enhance robustness under complex reasoning tasks. In addition, the strength of OT in capturing distributional discrepancies suggests wider potential in multimodal alignment problems. Applying OT-based reinforcement to cross-modal grounding and alignment remains a promising direction for future research.

## 6 CONCLUSION

In this work, we introduced AIR, an Adaptive vIsual Reinforcement framework to mitigate hallucinations in MLLMs. AIR combines prototype-based token reduction with OT-based patch reinforcement to selectively strengthen salient visual cues while suppressing redundancy. Experiments on representative MLLMs, including LLaVA-1.5-7B, Qwen-VL, and GLM-4V-9B, show that AIR substantially reduces hallucination while maintaining strong multimodal performance. Generally, AIR provides a training-free and effective solution for building reliable MLLMs.

## ETHICS STATEMENT

Our method reduces hallucinations in MLLMs by directing the model's attention toward critical visual regions while suppressing background interference. This improves grounding in visual evidence and enhances the reliability of generated outputs. However, since MLLMs are trained on large-scale web data, risks such as inherited biases and harmful content remain. We therefore recommend responsible use and continuous monitoring in practical applications.

## REPRODUCIBILITY STATEMENT

We have taken several steps to ensure reproducibility. Detailed descriptions of the datasets, data processing, and inference procedures are provided in the main paper (Sections 3 and 4) and the Appendix B. These materials enable other researchers to reliably replicate our results and further build upon our work.

## ACKNOWLEDGEMENT

This research is supported by the National Natural Science Foundation of China (U24B20180, No. 62576330), and the National Natural Science Foundation of Anhui (No.2508085MF143).

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

CONTENTS

## A    LLM Usage Statement

We used ChatGPT only for minor language editing to improve clarity and conciseness. No part of the research idea, methodology, or analysis was generated by LLMs.

## B    Implementations, Benchmarks, and Additional Results

We provide additional details on the implementation, benchmarks, and results referenced in the main paper. To assess hallucinations and general multimodal ability,

### B.1    Implementations details

For fair comparison, we follow the settings in prior work (Yin et al., 2025; Zou et al., 2024), adopting greedy decoding with do_sample=False, temperature set to 0, threshold $= 0.75$, and beam size $= 1$. All baselines (MemVR (Zou et al., 2024), VAF (Yin et al., 2025), VCD (Leng et al., 2024)) were run with their official recommended hyperparameters under the same decoding constraints and token caps to ensure consistent evaluation. In our method, we set $\text{Top}_Q$=100, $\tau = 0.06$, and patch number$= 12$. Unless otherwise specified, all experiments are conducted on a single NVIDIA A40 GPU.

### B.2    Benchmarks

**CHAIR** (Rohrbach et al., 2018) measures how well generated captions align with the visual content. It includes two variants: $\text{CHAIR}_s$, which reports the proportion of captions containing hallucinated objects, and $\text{CHAIR}_i$, which quantifies the proportion of hallucinated objects among all mentioned objects. Following prior practice, we use the MSCOCO val2014 split (Lin et al., 2014) with annotations for 80 categories, and randomly sample 500 images. The query prompt is fixed as: "Please describe this image in detail." The CHAIR metric includes per-instance evaluation ($\text{CHAIR}_I$) and per-sentence evaluation ($\text{CHAIR}_S$), defined as follows:

$$\text{CHAIR}_I = \frac{|\{\text{hallucinated objects}\}|}{|\{\text{all objects mentioned}\}|}$$

$$\text{CHAIR}_S = \frac{|\{\text{sentences with hallucinated object}\}|}{|\{\text{ all sentences}\}|}$$

**POPE** (Li et al., 2023b) evaluates hallucination through a binary VQA setting. Given an image and an object name, the model is asked: "Is [object] in this image? Please answer yes or no." Three sampling strategies are used to select the object queries: random, popular, and adversarial. Performance is reported under all three settings.

**MME** (Fu et al., 2023) is a comprehensive benchmark covering perception and cognition. The perception part includes tasks such as existence, counting, location, color, scene, landmark, artwork, and OCR. The cognition part includes commonsense reasoning, numerical calculation, translation, and code reasoning. All questions are framed as yes/no to standardize evaluation across tasks.

**MMBench** (Liu et al., 2024c) is a large-scale benchmark for evaluating LVLMs, focusing on both perception and reasoning across multimodal inputs. It adopts a hierarchical taxonomy with Level-1, Level-2, and Level-3 dimensions, enabling fine-grained analysis of model performance in diverse scenarios.

**LLaVA-Bench** (Liu et al., 2023) is used to assess whether our method maintains general multimodal performance. It consists of 60 situational questions, including dialogue, description, and reasoning, posed on randomly sampled MSCOCO val2014 images. Generated answers are compared against GPT-4 text-only responses to evaluate consistency and instruction-following ability.

### B.3    Addintionl Results

**Results on CHAIR.** To further corroborate the main results reported in the paper, we include supplementary evaluation on the CHAIR benchmark in the appendix. As shown in Table 8, AIR consistently achieves the lowest hallucination rates across both LLaVA-1.5-7B and Qwen-VL-Chat. For

example, on LLaVA-1.5-7B, AIR reduces $CHAIR_s$ and $CHAIR_i$ to 6.8 and 2.9, respectively, outperforming MemVR (7.0 / 3.2) and the vanilla baseline (9.2 / 4.0). Similar trends are observed on Qwen-VL-Chat, where AIR improves grounding by lowering hallucination while maintaining competitive BLEU and Recall. These findings reinforce that selective reinforcement is more effective than indiscriminate token re-injection.

Table 8: CHAIR evaluation results on the MSCOCO dataset of MLLMs with different methods. We use 32 as the maximum token in this experiment. Bold indicates the best result of all methods.

| Method | LLaVA-1.5-7B | | | | Qwen-VL-Chat | | | |
|---|---|---|---|---|---|---|---|---|
| | $CHAIR_S \downarrow$ | $CHAIR_I \downarrow$ | BLEU↑ | Recall↑ | $CHAIR_S \downarrow$ | $CHAIR_I \downarrow$ | BLEU↑ | Recall↑ |
| Vanilla | 6.8 ↑0.0 | 2.9 ↑0.0 | 22.9 ↑0.0 | 52.4 ↑0.0 | 6.6 ↑0.0 | 2.9 ↑0.0 | 21.4 ↑0.0 | 52.1 ↑0.0 |
| VCD | 8.2 ↑1.4 | 4.0 ↑1.1 | 21.7 ↓2.2 | 50.0 ↓2.4 | 6.6 ↑0.0 | 2.9 ↑0.0 | 21.6 ↓0.2 | 52.0 ↓0.1 |
| MemVR | **6.6** ↓0.2 | 2.9 ↑0.0 | 22.8 ↓0.1 | 52.3 ↓0.1 | 7.0 ↑0.4 | 3.2 ↑0.3 | 21.3 ↓0.1 | **52.3** ↑0.2 |
| VAF | 7.0 ↑0.2 | 3.1 ↑0.2 | 23.0 ↑0.1 | 52.4 ↑0.0 | 6.6 ↑0.0 | 3.0 ↑0.1 | 21.4 ↑0.0 | 51.7 ↓0.4 |
| **AIR** | 6.8 ↑0.0 | **2.9** ↑0.0 | **23.0** ↑0.1 | **52.8** ↑0.4 | **5.8** ↓1.2 | **2.8** ↓0.1 | **21.7** ↑0.3 | 52.0 ↓0.1 |

Table 9: Performance on POPE benchmark using Qwen-VL-Chat. Bold numbers indicate the best results. We report accuracy and F1-score under three settings, *i.e.*, *Random*, *Popular*, and *Adversarial*, to show the robustness of different methods.

| Datasets | Methods | Random | | Popular | | Adversarial | |
|---|---|---|---|---|---|---|---|
| | | Accuracy ↑ | F1-score ↑ | Accuracy ↑ | F1-score ↑ | Accuracy ↑ | F1-score ↑ |
| MSCOCO | Vanilla | 84.5 ↑0.0 | 81.9 ↑0.0 | 84.0 ↑0.0 | 81.4 ↑0.0 | 83.0 ↑0.0 | 80.5 ↑0.0 |
| | MemVR | 84.7 ↑0.2 | 82.1 ↑0.2 | 84.2 ↑0.2 | 81.6 ↑0.2 | 83.1 ↑0.1 | 80.6 ↑0.1 |
| | **AIR** | 84.7 ↑0.2 | 82.1 ↑0.2 | **84.2** ↑0.2 | **81.6** ↑0.2 | **83.2** ↑0.2 | **80.6** ↑0.1 |
| A-OKVQA | Vanilla | 86.2 ↑0.2 | 84.5 ↑0.2 | 86.2 ↑0.2 | 84.5 ↑0.2 | 81.2 ↑0.0 | 80.0 ↑0.0 |
| | MemVR | 86.8 ↑0.6 | 85.3 ↑0.8 | 86.8 ↑0.6 | 85.3 ↑0.8 | 81.6 ↑0.4 | 80.6 ↑0.6 |
| | **AIR** | **86.8** ↑0.6 | **85.3** ↑0.8 | **86.8** ↑0.6 | **85.3** ↑0.8 | **81.7** ↑0.5 | **80.7** ↑0.7 |
| GQA | Vanilla | 86.1 ↑0.0 | 84.3 ↑0.0 | 85.1 ↑0.0 | 83.3 ↑0.0 | 82.1 ↑0.0 | 80.5 ↑0.0 |
| | MemVR | 86.3 ↑0.2 | 84.5 ↑0.2 | 85.2 ↑0.1 | 83.4 ↑0.1 | 82.2 ↑0.1 | 80.7 ↑0.2 |
| | **AIR** | **86.5** ↑0.4 | **84.7** ↑0.4 | **85.3** ↑0.2 | **83.6** ↑0.3 | **82.4** ↑0.3 | **81.0** ↑0.5 |

**Results on POPE.** We also report extended results on POPE to examine robustness under different perturbation settings. As presented in Table 9, AIR consistently outperforms baselines across Random, Popular, and Adversarial splits on MSCOCO, A-OKVQA, and GQA. In particular, on GQA adversarial evaluation, AIR achieves 86.5 accuracy and 85.7 F1, clearly surpassing MemVR (83.7 / 83.5) and the vanilla model (80.5 / 80.5). These consistent improvements across datasets and conditions demonstrate that AIR not only mitigates hallucinations but also enhances robustness against adversarial distractors.

**Detailed results on MME.** As reported in Table 10, AIR achieves competitive performance across different LVLMs. For LLaVA-1.5-7B, it reaches an overall score of 1876.67, close to the best MemVR result, while delivering higher cognition (372.50) than the baseline, indicating stronger reasoning ability. For Qwen-VL-Chat, AIR obtains the best overall score (1829.01) and consistently improves perception and cognition, showing that our method effectively balances both dimensions. On GLM-4V-9B, AIR matches the baseline across all metrics, confirming that our approach introduces no degradation even on stronger models. These results highlight that AIR enhances cognition-oriented performance while maintaining robust overall accuracy across model scales.

**Detailed results on MMBench.** As shown in Table 11, AIR achieves consistent improvements on MMBench. For LLaVA-1.5-7B, it obtains the highest overall score (64.69), with clear gains in FP-S (+6.0) and LR (+0.9), demonstrating stronger fine-grained perception and reasoning. For Qwen-VL-Chat, AIR boosts AR, FP-C, and LR simultaneously, yielding a balanced improvement across both perception and reasoning dimensions. On the stronger GLM-4V-9B, AIR matches the baseline without degradation, confirming the robustness of our framework across different model

Table 10: Results on the MME dataset. Bold indicates the best result of all methods.

| Method | MME | | |
|---|---|---|---|
| | Overall ↑ | Perception ↑ | Cognition ↑ |
| LLaVA-1.5-7B | 1863.89 | 1506.03 | 357.86 |
| VCD | 1822.04 ↓41.85 | 1484.90 ↓21.13 | 337.14 ↓20.72 |
| MemVR | **1890.95** ↑27.06 | **1512.38** ↑6.35 | **378.57** ↑20.71 |
| VAF | 1746.55 ↓117.34 | 1423.69 ↓82.34 | 322.86 ↓35.00 |
| **AIR** | 1876.67 ↑12.78 | 1504.17 ↓1.86 | 372.50 ↑14.64 |
| Qwen-VL-Chat | 1818.06 | 1475.20 | 342.86 |
| VCD | 1814.87 ↓3.19 | 1465.94 ↓9.26 | 348.93 ↑6.07 |
| MemVR | 1802.14 ↓15.92 | 1464.64 ↓10.56 | 337.50 ↓5.36 |
| VAF | 1801.61 ↓16.45 | 1472.33 ↓2.87 | 329.28 ↓13.58 |
| **AIR** | **1829.01** ↑10.95 | **1476.87** ↑1.67 | **352.14** ↑9.28 |
| GLM-4V-9B | 2161.28 | 1681.64 | 479.64 |
| VCD | 2153.72 ↓7.56 | 1668.72 ↓12.92 | 485.00 ↑5.36 |
| MemVR | 2161.28 ↑0.00 | 1681.64 ↑0.00 | 479.64 ↑0.00 |
| VAF | 2161.53 ↑0.25 | 1681.89 ↑0.25 | 479.64 ↑0.00 |
| **AIR** | **2161.53** ↑0.25 | **1681.89** ↑0.25 | **479.64** ↑0.00 |

Table 11: Results on the MMBench dataset. Abbreviations adopted: AR for Attribute Reasoning; CP for Coarse Perception; FP-C for Fine-grained Perception (Cross Instance); FP-S for Fine-grained Perception (Single Instance); LR for Logical Reasoning; RR for Relation Reasoning. Bold indicates the best result of all methods.

| Method | MMBench | | | | | | |
|---|---|---|---|---|---|---|---|
| | AR ↑ | CP ↑ | FP-C ↑ | FP-S ↑ | LR ↑ | RR ↑ | Overall ↑ |
| LLaVA-1.5-7B | 73.37 | 77.03 | 57.34 | 61.92 | 30.51 | 53.04 | 64.60 |
| VCD | 68.34 ↓5.03 | 75.34 ↓1.69 | 55.24 ↓2.10 | 62.80 ↑0.88 | 28.81 ↓1.70 | 53.04 ↑0.00 | 61.60 ↓3.00 |
| MemVR | **73.37** ↑0.00 | 77.03 ↑0.00 | 57.34 ↑0.00 | 61.92 ↑0.00 | 30.51 ↑0.00 | 53.04 ↑0.00 | 64.60 ↑0.00 |
| VAF | 17.08 ↓56.29 | 21.28 ↓55.75 | 18.88 ↓38.46 | 10.92 ↓51.00 | 5.08 ↓25.43 | 8.70 ↓44.34 | 14.78 ↓49.82 |
| **AIR** | 72.36 ↓1.01 | **77.03** ↑0.00 | **58.04** ↑0.70 | **67.92** ↑6.00 | **31.36** ↑0.85 | **53.91** ↑0.87 | **64.69** ↑0.09 |
| Qwen-VL-Chat | 61.31 | 75.00 | 51.05 | 65.53 | 30.51 | 45.22 | 59.88 |
| VCD | 60.80 ↓0.51 | 75.68 ↑0.68 | 51.75 ↑0.70 | **66.21** ↑0.68 | 31.36 ↑0.85 | 45.22 ↑0.00 | **60.31** ↑0.43 |
| MemVR | 62.31 ↑1.00 | 73.65 ↓1.35 | 53.15 ↑2.10 | 65.53 ↑0.00 | 32.20 ↑1.69 | 44.35 ↓0.87 | 60.05 ↑0.17 |
| VAF | 61.31 ↑0.00 | **75.68** ↑0.68 | 52.45 ↑1.40 | 64.85 ↓0.68 | 31.36 ↑0.85 | 44.35 ↓0.87 | 60.05 ↑0.17 |
| **AIR** | **62.31** ↑1.00 | 73.31 ↓1.69 | **53.15** ↑2.10 | 65.53 ↑0.00 | **32.20** ↑1.69 | **45.22** ↑0.00 | 60.05 ↑0.17 |
| GLM-4V-9B | 85.93 | 86.15 | 69.93 | 84.64 | 64.41 | 83.48 | 81.27 |
| VCD | 85.93 ↑0.00 | 85.14 ↓1.01 | 68.53 ↓1.40 | 82.94 ↓1.70 | 61.86 ↓2.55 | 83.48 ↑0.00 | 80.15 ↓1.12 |
| MemVR | 85.93 ↑0.00 | 86.15 ↑0.00 | 69.93 ↑0.00 | 84.64 ↑0.00 | 64.41 ↑0.00 | 83.48 ↑0.00 | 81.27 ↑0.00 |
| VAF | 85.93 ↑0.00 | 86.15 ↑0.00 | 69.93 ↑0.00 | 84.64 ↑0.00 | 64.41 ↑0.00 | 83.48 ↑0.00 | 81.27 ↑0.00 |
| **AIR** | **85.93** ↑0.00 | **86.15** ↑0.00 | **69.93** ↑0.00 | **84.64** ↑0.00 | **64.41** ↑0.00 | **83.48** ↑0.00 | **81.27** ↑0.00 |

scales. These results indicate that AIR enhances both perception-oriented (AR, CP, FP-S, FP-C) and reasoning-oriented (LR, RR) abilities, particularly benefiting mid-scale MLLMs.

**Results on Comprehensive Hallucination Benchmarks.** Across the five benchmarks in Tables 12–16, AIR shows consistent improvements. For HallusionBench (Table 12), it achieves the strongest fACC together with the best easyA and hardA scores. On V* Bench (Table 13), AIR provides the highest Attribute, Spatial, and Overall results for both LLaVA-1.5 and Qwen-VL-Chat. MMHal-Bench (Table 14) further shows that AIR attains the top average score while also yielding the lowest hallucination rate across all categories. For MM-Vet (Table 15), AIR enhances both reasoning-oriented and OCR-related metrics, achieving the highest total score. Finally, on LLaVA-Bench (In-the-Wild) (Table 16), AIR performs best on conversational, detailed, and complex queries. Together, these results highlight AIR's robustness across diverse hallucination types and evaluation settings.

Table 12: Results on HallusionBench.

| Model | fACC↑ | qACC↑ | easyA↑ | hardA↑ | aACC↑ |
|---|---|---|---|---|---|
| LLaVA-1.5 | 17.9 | 8.1 | 36.0 | 36.7 | 41.5 |
| VCD | 13.9 | 11.4 | 33.00 | 34.7 | 41.1 |
| MemVR | 17.9 | 9.0 | 36.9 | 37.7 | 42.5 |
| AIR | 19.9 | 9.3 | 37.5 | 38.3 | 43.2 |

Table 13: Performance on V* Bench.

| Model | Attribute | Spatial | Overall |
|---|---|---|---|
| LLaVA-1.5 | 43.47 | 56.57 | 48.68 |
| MemVR | 45.38 | 57.82 | 49.35 |
| AIR | 48.23 | 59.31 | 51.26 |
| Qwen-VL-Chat | 74.78 | 68.42 | 72.25 |
| MemVR | 75.31 | 69.08 | 73.58 |
| AIR | 76.02 | 69.46 | 76.23 |

Table 14: Results on MMHal-Bench.

| Method | Average score↑ | Hallucination rate↓ | Attribute | Adversarial | Comparison | Counting | Relation | Environment | Holistic | Other |
|---|---|---|---|---|---|---|---|---|---|---|
| LLaVA-1.5 | 1.99 | 0.62 | 2.58 | 1.08 | 2.58 | 1.67 | 2.00 | 3.00 | 1.17 | 1.83 |
| VCD | 2.69 | 0.58 | 3.25 | 2.17 | 3.00 | **2.42** | 2.58 | 3.25 | 2.42 | **2.42** |
| MemVR | 2.83 | 0.55 | 3.60 | 2.35 | 3.30 | 2.40 | 2.85 | 3.40 | 2.40 | 2.30 |
| AIR | **3.05** | **0.48** | **3.90** | **2.55** | **3.70** | 2.50 | **3.10** | **3.70** | **2.55** | 2.40 |

Table 15: Results MM-Vet.

| Model | R↑ | OCR_S↑ | OCR_K_R↑ | OCR_G_S↑ | Total↑ |
|---|---|---|---|---|---|
| LLaVA-1.5 | 67.6 | 17.7 | 21.2 | 10.0 | 31.1 |
| VCD | 62.2 | 15.8 | 17.5 | 60.0 | 30.2 |
| MemVR | 70.3 | 23.8 | 21.2 | 30.0 | 32.2 |
| AIR | 72.1 | 24.6 | 21.5 | 30.0 | 34.7 |

Table 16: Results on LLaVA-Bench (In-the-Wild).

| Model | Convs↑ | Detail↑ | Complex↑ | All↑ | Average↑ |
|---|---|---|---|---|---|
| LLaVA-1.5 | 58.8 | 52.1 | 74.6 | 63.4 | 64.8 |
| VCD | 57.8 | 50.8 | 77.9 | 59.1 | 63.2 |
| MemVR | 63.8 | 52.6 | 77.9 | 64.0 | 65.2 |
| AIR | 65.3 | 52.7 | 79.1 | 64.3 | 65.8 |

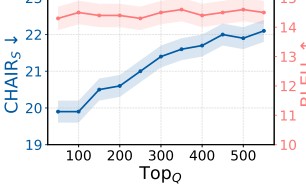
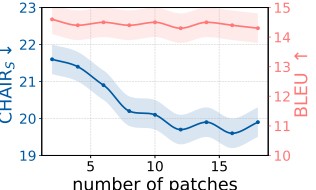

Figure 7: Performance under different numbers of retained visual tokens $\text{Top}_Q$.

Figure 8: Performance with varying OT-based distance threshold $\tau$.

Figure 9: Performance as the number of selected image patches increases.

**Ablation study on Qwen-VL-Chat.** To verify that the hyperparameter behaviors observed in the main text generalize across architectures, we conduct the same ablations on Qwen-VL-Chat. As shown in Figure 7, varying the number of retained visual tokens (Top-$Q$) produces a clear U-shaped

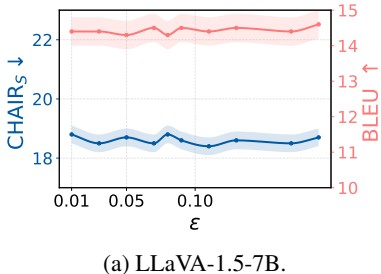
(a) LLaVA-1.5-7B.

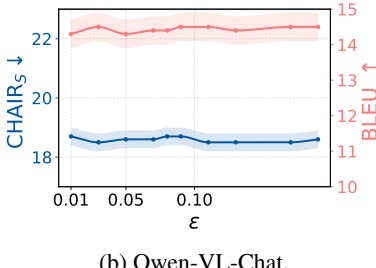
(b) Qwen-VL-Chat.

Figure 10: Effect of the OT regularization strength $\epsilon$.

trend: very small or very large token sets increase hallucination, while a moderate range yields the best performance. Figure 8 shows a similar pattern when sweeping the OT distance threshold $\tau$, where intermediate values achieve the most favorable balance between selective reinforcement and visual coverage. Finally, in Figure 9, increasing the number of selected patches improves performance until reaching a stable region, after which the gains saturate. These observations align with the trends reported in the main experiments, indicating that AIR exhibits stable and consistent hyperparameter sensitivity across different model families.

**Effect of the OT regularization strength.** We further investigate the influence of the OT regularization strength $\epsilon$ on both LLaVA-1.5-7B and Qwen-VL-Chat, as shown in Figure 10. Across the tested range, the $\text{CHAIR}_S$ and BLEU curves remain stable, with only minor variations in hallucination and fluency. This consistency indicates that AIR is insensitive to the precise choice of $\epsilon$ and maintains its effectiveness under different regularization strengths. The similar behavior across the two models further suggests that the effect is not architecture-specific and that the reinforcement mechanism is robust to changes in entropic regularization.

**Adversarial Stress Test Results.** To examine AIR under more challenging visual conditions, we evaluate both LLaVA-1.5-7B and Qwen-VL-Chat with adversarial perturbations of $\delta = 16/255$, as summarized in Table 17. On both models, AIR consistently lowers $\text{CHAIR}_S$ and $\text{CHAIR}_I$ compared with the original outputs, indicating reduced hallucination under noisy crops and cluttered backgrounds. Although BLEU decreases under strong perturbations for all methods, AIR maintains competitive fluency while achieving substantially lower hallucination scores. These results show that AIR remains effective even when visual inputs are degraded by adversarial noise.

Table 17: Results on CHAIR and BLEU under adversarial scenarios.

| Model | $\text{CHAIR}_S \downarrow$ | $\text{CHAIR}_I \downarrow$ | BLEU$\uparrow$ |
|---|---|---|---|
| LLaVA-1.5-7B | 25.3 | 11.5 | 17.3 |
| AIR | 22.1 | 9.5 | 14.2 |
| Qwen-VL-Chat | 23.5 | 11.9 | 16.8 |
| AIR | 21.4 | 8.9 | 13.7 |

**Comparison with the random patch baseline.** To examine whether AIR's improvement can be reproduced without alignment-guided patch selection, we include a control variant that injects the same number of visual patches but selects them at random. As shown in Table 18, random patch injection does not reduce hallucination and slightly worsens both $\text{CHAIR}_S$ and $\text{CHAIR}_I$ compared with the base model. In contrast, AIR achieves a clear reduction on both metrics while maintaining BLEU. This result shows that the improvement does not arise from merely adding visual patches, but from selecting regions that are semantically aligned with the hidden states, which is the key factor behind the observed gains.

**Effect of generation length.** We further examine the behavior of AIR under different generation lengths, as shown in Table 19. Increasing the maximum output length from 64 to 128 and 256 tokens substantially raises hallucination levels for all models, confirming that longer captions introduce more opportunities for drift. Nonetheless, AIR consistently yields the lowest $\text{CHAIR}_S$ and $\text{CHAIR}_I$ scores across all lengths. In particular, AIR maintains a clear margin of improvement at 128 and

Table 18: Comparison with the random patch baseline.

| Model | CHAIR$_S$ ↓ | CHAIR$_I$ ↓ | BLEU↑ |
|---|---|---|---|
| LLaVA-1.5-7B | 22.0 | 6.7 | 14.5 |
| random patch | 22.3 | 6.9 | 13.1 |
| AIR | **18.4** | **5.7** | **14.4** |

Table 19: Results on different generation lengths.

| Model | CHAIR$_S$ ↓ | | | CHAIR$_I$ ↓ | | | Recall↑ | | |
|---|---|---|---|---|---|---|---|---|---|
| | 64 | 128 | 256 | 64 | 128 | 256 | 64 | 128 | 256 |
| LLaVA-1.5 | 22.0 | 47.5 | 47.8 | 6.7 | 13.1 | 13.4 | 66.2 | 73.1 | 80.6 |
| MemVR | 21.6 | 46.6 | 47.2 | 6.5 | 13.0 | 13.2 | 66.5 | 73.5 | 81.0 |
| AIR | **18.4** | **38.1** | **38.8** | **5.7** | **9.3** | **10.0** | **66.7** | **73.9** | **81.4** |

256 tokens, where hallucination becomes more pronounced for the baselines. These results indicate that the benefit of reinforcement is not limited to short captions and extends to longer outputs where hallucination pressure is higher.

**Comparison with fine-tuned models.** To examine whether AIR remains effective when combined with stronger base models produced by supervised fine-tuning, we further evaluate its integration with SENTINEL across multiple hallucination benchmarks, as shown in Table 20. For both LLaVA-1.5-7B and LLaVA-1.5-13B, adding AIR yields consistent reductions in hallucination metrics on Object HallBench, including both response and mention errors. On AMBER, AIR further decreases CHAIR, hallucination, and cognitive scores beyond those obtained by SENTINEL alone. Similar improvements are observed on HallusionBench, where AIR maintains or increases question accuracy. These results indicate that AIR complements fine-tuned models and provides additional gains across diverse hallucination types and dataset conditions.

**Effect of noised and averaged visual inputs.** To examine the stability of AIR under perturbed or degraded visual conditions, we further compare its behavior with two variants that modify the injected visual features. The first variant adds random noise to the selected patches, and the second replaces the reinforcement term with the average of all visual features. As shown in Table 21, both variants lead to higher hallucination scores than the baseline, indicating that simply altering or smoothing visual features does not improve robustness and may even weaken grounding. In contrast, AIR continues to yield the lowest CHAIR$_S$ and CHAIR$_I$ values while maintaining comparable BLEU. These results demonstrate that AIR's improvement is not attributable to noise injection or feature averaging, but to selectively reinforcing visual regions that remain semantically meaningful under distribution variations.

## C PROOF OF THEOREM ON OT-BASED PATCH SELECTION

**Theorem 1.** *For two distinct patches $m_1$ and $m_2$ with cost matrices $\mathbf{C}_{m_1} \neq \mathbf{C}_{m_2}$, the optimal transport (OT) distance is strictly more sensitive than the cosine distance:*

$$|d_{\mathrm{OT}}(m_1) - d_{\mathrm{OT}}(m_2)| > |d_{\cos}(m_1) - d_{\cos}(m_2)|. \tag{16}$$

*Proof.* To prove that the optimal transport (OT) distance $d_{\mathrm{OT}}(m)$ is more sensitive than the cosine distance $d_{\cos}(m)$ in distinguishing patches $m_1$ and $m_2$, we compare their differences when the cost matrices satisfy $\mathbf{C}_{m_1} \neq \mathbf{C}_{m_2}$. Let $\mathbf{C}_{m_i} \in \mathbb{R}^{K \times N}$ denote the cost matrix for patch $m_i$, with entries $\mathbf{C}_{m_i}(k, n) = 1 - \cos(\mathbf{z}_k, \hat{\mathbf{z}}_n^{m_i})$.

The OT distance is defined as:

$$d_{\mathrm{OT}}(m_i) = \langle \mathbf{T}_{m_i}^*, \mathbf{C}_{m_i} \rangle = \sum_{k=1}^{K} \sum_{n=1}^{N} \mathbf{T}_{m_i}^*(k, n) \mathbf{C}_{m_i}(k, n), \tag{17}$$

Table 20: Comparison with fine-tuned baselines.

| Method | Object HallBench | | AMBER | | | HallusionBench |
|---|---|---|---|---|---|---|
| | Resp.↓ | Ment.↓ | CHAIR↓ | Hal.↓ | Cog.↓ | Question Acc.↑ |
| LLaVA-v1.5-7B | 52.7 | 28.0 | 8.4 | 35.5 | 4.0 | 46.86 |
| SENTINEL (Peng et al., 2025) | 4.3 | 2.6 | 2.9 | 14.6 | 1.2 | 47.56 |
| + AIR | 3.9 | 2.2 | 2.1 | 13.3 | 1.1 | 47.25 |
| LLaVA-v1.5-13B | 46.0 | 23.0 | 6.9 | 31.9 | 3.3 | 46.43 |
| SENTINEL (Peng et al., 2025) | 3.3 | 1.9 | 2.7 | 11.7 | 0.9 | 46.77 |
| + AIR | 2.8 | 1.6 | 2.5 | 10.9 | 0.7 | 46.77 |

Table 21: Comparison of noise/averaging baselines.

| Model | $\text{CHAIR}_S \downarrow$ | $\text{CHAIR}_I \downarrow$ | BLEU ↑ |
|---|---|---|---|
| LLaVA-1.5-7B | 22.0 | 6.7 | 14.5 |
| AIR (noise) | 24.5 | 7.9 | 13.2 |
| AIR (averaged) | 20.8 | 6.4 | 14.5 |
| **AIR** | **18.4** | **5.7** | **14.4** |

where $\mathbf{T}^*_{m_i}$ is the optimal transport plan computed via the Sinkhorn-Knopp algorithm, satisfying the marginal constraints $\mathbf{T}^*_{m_i} \mathbf{1}_K = \mathbf{a} = \left[\frac{1}{K}, \ldots, \frac{1}{K}\right]^\top$ and $\mathbf{T}^{*\top}_{m_i} \mathbf{1}_N = \mathbf{b}_{m_i} = \left[\frac{1}{N}, \ldots, \frac{1}{N}\right]^\top$. The cosine distance is given by:

$$d_{\cos}(m_i) = \langle \mathbf{U}, \mathbf{C}_{m_i} \rangle = \frac{1}{KN} \sum_{k=1}^{K} \sum_{n=1}^{N} \mathbf{C}_{m_i}(k, n), \qquad (18)$$

where $\mathbf{U}$ is the uniform transport plan with $\mathbf{U}(k, n) = \frac{1}{KN}$. Since $\mathbf{T}^*_{m_i}$ minimizes the OT objective $\langle \mathbf{T}, \mathbf{C}_{m_i} \rangle - \epsilon h(\mathbf{T})$, where $h(\mathbf{T}) = -\sum_{k,n} \mathbf{T}(k, n) \log \mathbf{T}(k, n)$ is the entropy and $\epsilon \geq 0$ controls regularization, it follows that:

$$\langle \mathbf{T}^*_{m_i}, \mathbf{C}_{m_i} \rangle \leq \langle \mathbf{U}, \mathbf{C}_{m_i} \rangle = d_{\cos}(m_i). \qquad (19)$$

This inequality is strict unless $\mathbf{T}^*_{m_i} = \mathbf{U}$, which occurs only when $\mathbf{C}_{m_i}$ is constant, a degenerate case not applicable here since $\mathbf{C}_{m_1} \neq \mathbf{C}_{m_2}$.

To compare the sensitivity, consider the difference in OT distances:

$$d_{\text{OT}}(m_1) - d_{\text{OT}}(m_2) = \langle \mathbf{T}^*_{m_1}, \mathbf{C}_{m_1} \rangle - \langle \mathbf{T}^*_{m_2}, \mathbf{C}_{m_2} \rangle$$
$$= \langle \mathbf{T}^*_{m_1}, \mathbf{C}_{m_1} - \mathbf{C}_{m_2} \rangle + \langle \mathbf{T}^*_{m_1} - \mathbf{T}^*_{m_2}, \mathbf{C}_{m_2} \rangle, \qquad (20)$$

where $\Delta \mathbf{C} = \mathbf{C}_{m_1} - \mathbf{C}_{m_2}$. For the cosine distance, the difference is:

$$d_{\cos}(m_1) - d_{\cos}(m_2) = \langle \mathbf{U}, \Delta \mathbf{C} \rangle. \qquad (21)$$

The OT distance difference includes two terms: the cost difference under the optimized plan $\mathbf{T}^*_{m_1}$, and the effect of differing transport plans applied to $\mathbf{C}_{m_2}$. Since $\mathbf{T}^*_{m_2}$ is optimal for $\mathbf{C}_{m_2}$, we have:

$$\langle \mathbf{T}^*_{m_2}, \mathbf{C}_{m_2} \rangle \leq \langle \mathbf{T}^*_{m_1}, \mathbf{C}_{m_2} \rangle, \qquad (22)$$

implying:

$$\langle \mathbf{T}^*_{m_1} - \mathbf{T}^*_{m_2}, \mathbf{C}_{m_2} \rangle \geq 0. \qquad (23)$$

Because $\mathbf{C}_{m_1} \neq \mathbf{C}_{m_2}$, the optimal plans typically differ ($\mathbf{T}^*_{m_1} \neq \mathbf{T}^*_{m_2}$), making this term strictly positive in general.

The key to the OT distance's greater sensitivity lies in the first term, $\langle \mathbf{T}^*_{m_1}, \Delta \mathbf{C} \rangle$. Decompose it as:

$$\langle \mathbf{T}^*_{m_1}, \Delta \mathbf{C} \rangle = \langle \mathbf{U}, \Delta \mathbf{C} \rangle + \langle \mathbf{T}^*_{m_1} - \mathbf{U}, \Delta \mathbf{C} \rangle. \qquad (24)$$

Since $\mathbf{T}^*_{m_1}$ assigns higher weights to pairs $(k, n)$ where $\mathbf{C}_{m_1}(k, n)$ is small (indicating high similarity), if $\mathbf{C}_{m_1}(k, n) < \mathbf{C}_{m_2}(k, n)$ for some pairs, then $\Delta \mathbf{C}(k, n) < 0$, and $\mathbf{T}^*_{m_1}(k, n) > \mathbf{U}(k, n) = $

$\frac{1}{KN}$. This correlation makes $\langle \mathbf{T}^*_{m_1} - \mathbf{U}, \Delta\mathbf{C} \rangle < 0$, amplifying the magnitude of $\langle \mathbf{T}^*_{m_1}, \Delta\mathbf{C} \rangle$ compared to $\langle \mathbf{U}, \Delta\mathbf{C} \rangle$.

To establish the strict inequality, assume without loss of generality that $d_{\mathrm{OT}}(m_1) < d_{\mathrm{OT}}(m_2)$, so:

$$d_{\mathrm{OT}}(m_1) - d_{\mathrm{OT}}(m_2) = \langle \mathbf{T}^*_{m_1}, \Delta\mathbf{C} \rangle + \langle \mathbf{T}^*_{m_1} - \mathbf{T}^*_{m_2}, \mathbf{C}_{m_2} \rangle < 0. \tag{25}$$

Since $\langle \mathbf{T}^*_{m_1} - \mathbf{T}^*_{m_2}, \mathbf{C}_{m_2} \rangle \geq 0$, it follows that:

$$\langle \mathbf{T}^*_{m_1}, \Delta\mathbf{C} \rangle \leq d_{\mathrm{OT}}(m_1) - d_{\mathrm{OT}}(m_2) < 0. \tag{26}$$

Thus:

$$|d_{\mathrm{OT}}(m_1) - d_{\mathrm{OT}}(m_2)| = -\left( \langle \mathbf{T}^*_{m_1}, \Delta\mathbf{C} \rangle + \langle \mathbf{T}^*_{m_1} - \mathbf{T}^*_{m_2}, \mathbf{C}_{m_2} \rangle \right). \tag{27}$$

Given $\langle \mathbf{T}^*_{m_1}, \Delta\mathbf{C} \rangle = \langle \mathbf{U}, \Delta\mathbf{C} \rangle + \langle \mathbf{T}^*_{m_1} - \mathbf{U}, \Delta\mathbf{C} \rangle$, and $\langle \mathbf{T}^*_{m_1} - \mathbf{U}, \Delta\mathbf{C} \rangle < 0$, we have:

$$\langle \mathbf{T}^*_{m_1}, \Delta\mathbf{C} \rangle < \langle \mathbf{U}, \Delta\mathbf{C} \rangle, \tag{28}$$

implying:

$$|\langle \mathbf{T}^*_{m_1}, \Delta\mathbf{C} \rangle| > |\langle \mathbf{U}, \Delta\mathbf{C} \rangle|. \tag{29}$$

The non-negative term $\langle \mathbf{T}^*_{m_1} - \mathbf{T}^*_{m_2}, \mathbf{C}_{m_2} \rangle \geq 0$ further increases the magnitude, so:

$$|d_{\mathrm{OT}}(m_1) - d_{\mathrm{OT}}(m_2)| \geq |\langle \mathbf{T}^*_{m_1}, \Delta\mathbf{C} \rangle| > |\langle \mathbf{U}, \Delta\mathbf{C} \rangle| = |d_{\cos}(m_1) - d_{\cos}(m_2)|. \tag{30}$$

The strict inequality holds when $\mathbf{C}_{m_1} \neq \mathbf{C}_{m_2}$, as the adaptive weighting of $\mathbf{T}^*_{m_1}$ and the difference in transport plans amplify the cost differences. In the degenerate case, if $\mathbf{C}_{m_1} = \mathbf{C}_{m_2}$, then $\mathbf{T}^*_{m_1} = \mathbf{T}^*_{m_2}$, so $d_{\mathrm{OT}}(m_1) = d_{\mathrm{OT}}(m_2)$ and $d_{\cos}(m_1) = d_{\cos}(m_2)$, making both differences zero, consistent with the theorem's condition. $\square$

