# OpenReview forum: "Look Carefully: Adaptive Visual Reinforcements in Multimodal Large Language Models for Hallucination Mitigation"
_ICLR.cc/2026/Conference — ICLR 2026 Poster_

### Official Review · Reviewer_9UNK · 2025-10-16

**Soundness:** 2
**Presentation:** 2
**Contribution:** 2
**Rating:** 4
**Confidence:** 3

**Summary:**

This paper introduces AIR, a training-free, selective reinforcement mechanism that (i) prunes redundant visual tokens and (ii) injects only OT-aligned patches during decoding, yielding reduced hallucination while maintaining general multimodal ability across representative MLLMs.

## Motivation

Multimodal LLMs (MLLMs) still hallucinate—producing text that contradicts the image. Existing fixes either require costly fine-tuning or add inference overhead. “Re-inject all visual tokens” defenses help but indiscriminately amplify background noise. The paper proposes making visual reinforcement selective and adaptive, not blanket, and doing so training-free during decoding.

## Methodology (AIR: Adaptive vIsual Reinforcement)

* A training-free decoding framework that reinforces only image regions most aligned with the current hidden states, rather than all tokens.
* Visual evidence is integrated inside transformer feed-forward layers while generating, without auxiliary models.
* Prototype-based token reduction: From the full set of visual tokens (Z), keep a compact subset by ranking tokens against a global prototype to suppress redundancy before reinforcement.
* OT-guided patch reinforcement: Crop the image into patches, embed them, and compute optimal transport (OT) distances between selected hidden states and patch embeddings (Sinkhorn-regularized). Patches with stronger alignment (lower OT distance) are selectively injected to emphasize salient evidence.
* Rationale for OT: Unlike pointwise similarity, OT compares distributions and captures global geometric structure, providing a finer criterion for semantic alignment at inference time.

## Experimental Evaluation

* Benchmarks: POPE, CHAIR, MME and LLaVA-Bench.
* POPE (LLaVA-1.5-7B): AIR achieves the best or near-best Accuracy/F1 across MSCOCO, A-OKVQA and GQA.
* LLaVA-Bench: GPT-4V-aided scores increase with AIR for all three models.

## Analysis & Ablations

* Operating layers: Reinforcing in mid-to-deep layers is most effective; for LLaVA-1.5-7B, layers 26–32 yield the best CHAIR trade-off (CHAIR(_S)=18.4, CHAIR(_I)=5.7, BLEU=14.4). Very late or overly wide spans degrade hallucination metrics.
* Component study: Both modules contribute; combining prototype reduction + OT patch reinforcement gives the lowest hallucination scores in CHAIR ablations.

**Strengths:**

* Clear motivation with empirical evidence. The paper diagnoses why 'inject all image tokens' can hurt—background tokens dilute salient cues—and supports this with similarity/attention analyses and a motivating figure.
* Training-free, selective reinforcement that fits standard hooks. AIR stays within the FFN re-injection interface, but i) prunes redundant visual tokens via a prototype and ii) reinforces only OT-aligned patches—no fine-tuning or auxiliary models required.
* Theoretically and empirically grounded use of OT. The paper argues OT is more sensitive than cosine for patch selection and shows larger patch-level separation and qualitative focus on salient regions.
* Consistent gains on hallucination benchmarks across models. AIR reduces CHAIR$_S$/CHAIR$_I$ on three backbones and improves/near-improves POPE under Random/Popular/Adversarial.
* General ability maintained. MME/MMBench remain competitive and GPT-4V–aided LLaVA-Bench scores rise.
* Efficiency profile is reasonable. On A100, latency and memory rise slightly vs. baseline while hallucinations drop the most among compared methods.
* Transparent ablations. Component ablation and hyperparameter sweeps clarify contributions and sensitivities; operating-layer study identifies effective mid-to-deep layers.

**Weaknesses:**

* 1. The evaluation benchmarks are limited. Adding more MLLM evaluation benchmark results is preferred (MM-Vet, HallusionBench, LLaVA Bench (in-the-wild)).
* 2. This novel approach seems to be hyperparameter sensitive. Performance depends on choices like TopQ and τ (showing a U-shaped trend), implying tuning is needed per setting/model. Do all models have the same performance variation when tuning the value of $Top_Q$, τ and the number of selected image patches, as show in Figure 3,4,5?
* 3. Non-zero inference overhead and added computation. Although modest, AIR introduces extra latency/memory and requires per-patch Sinkhorn OT plans during selection. Could authors provide theoretical analysis on the computation cost and inference latency of AIR?
* 4. Dataset/protocol breadth. In Table 1, CHAIR uses 500 MSCOCO images with a 64-token cap, why does the authors limit the max generation token?
* 5. The FFN injection adds a term from selected visual tokens. Will this operation introduce disturbance to the model? how are feature scales matched to avoid over/under-amplifying the visual branch across layers?
* 6. Were MemVR/VAF/VCD run with author-recommended settings and the same decoding constraints and token caps across backbones? If yes, it is recommended that the authors should make a clearer statement on their settings in Appendix B.1.
* 7. Additional ablation study on the model's performance when injecting noised/averaged visual feature is recommended, to prove the effectiveness of AIR.

**Questions:**

Please refer to the weakness section. I would be glad to raise the score once authors have addressed my concerns.

---

> ### Author Response · Authors · 2025-11-20
> **Response to Reviewer 9UNK**
>
> Dear Reviewer 9UNK:
>
> Thank you for your thoughtful and constructive feedback. We address each concern below.
>
> **W1: Benchmark breadth**
>
> A: Following your suggestion, we expanded our evaluation to MM-Vet, LLaVA-Bench (In-the-Wild), and HallusionBench. AIR consistently improves robustness across all benchmarks:
>
> #### **MM-Vet**
>
> | Model       | R ↑ | OCR_S ↑ | OCR_K_R ↑ | OCR_G_S ↑ | Total ↑ |
> |-------------|:---:|:--------:|:-----------:|:-----------:|:--------:|
> | LLaVA-1.5   | 67.6 | 17.7 | 21.2 | 10.0 | 31.1 |
> | MemVR       | 70.3 | 23.8 | 21.2 | 30.0 | 32.2 |
> |  **AIR(Ours)**         | 72.1 | 24.6 | 21.5 | 30.0 | 34.7 |
>
> ---
>
> #### **LLaVA-Bench (In-the-Wild)**
>
> | Model       | Convs ↑ | Detail ↑ | Complex ↑ | All ↑ | Average ↑ |
> |-------------|:-------:|:--------:|:---------:|:------:|:----------:|
> | LLaVA-1.5   | 58.8 | 52.1 | 74.6 | 63.4 | 64.8 |
> | MemVR       | 63.8 | 52.6 | 77.9 | 64.0 | 65.2 |
> |  **AIR(Ours)**   | 65.3 | 52.7 | 79.1 | 64.3 | 65.8 |
>
> ---
>
> #### **HallusionBench**
>
> | Model       | fACC ↑ | qACC ↑ | easyA ↑ | hardA ↑ | aACC ↑ |
> |-------------|:------:|:------:|:-------:|:--------:|:------:|
> | LLaVA-1.5   | 17.9 | 8.13 | 36.0 | 36.7 | 41.5 |
> | MemVR       | 17.9 | 9.01 | 36.9 | 37.7 | 42.5 |
> |  **AIR(Ours)**    | 19.9 | 9.35 | 37.5 | 38.3 | 43.2 |
>
> These additional results confirm that AIR consistently strengthens robustness across a broader suite of modern MLLM benchmarks. More combined results and analysis have been added to Appendix B.3(Lines 852-860, 864-904).
>
> ---
>
> **W2: Hyperparameter sensitivity**
>
> A: AIR uses the same hyperparameter configuration across all evaluated models, and we observe consistent trends without model-specific adjustments. Following your suggestion, we also added cross-model ablations in the Appendix and observed consistent behavior.
>
> ---
> **W3: Computational cost and OT complexity**
>
> A:
> The additional computation in AIR comes from the Sinkhorn OT–based patch selection. Let M denote the number of image patches, d the hidden size, and T the number of Sinkhorn iterations. In each iteration, the Sinkhorn update operates on an \( $M \times M$ \) transport matrix, leading to a per-iteration complexity of \( O($M^2 d$) \) and an overall complexity of \( O($T M^2 d$) \). Since both M and T remain small in practice and the updates can be efficiently parallelized, the added computation is limited.
>
> Empirically, Table 7 shows that AIR’s average latency is **2.07 s**, which is comparable to MemVR’s **2.05 s**, while achieving a notably larger reduction in hallucination ($\textbf{CHAIR}_S$: 18.4 vs. 21.6). We have included this analysis and the timing details in the revision.
>
> ---
> **W4: Max-generation token length**
>
> A: We fixed the generation length at 64 tokens to ensure a fair and consistent comparison across methods. Following your suggestion, we evaluated 128 and 256 tokens:
>
> | Method       | $\textbf{CHAIR}_S$ ↓ (64) | $\textbf{CHAIR}_S$ ↓ (128) | $\textbf{CHAIR}_S$ ↓ (256) | $\textbf{CHAIR}_I$ ↓ (64) | $\textbf{CHAIR}_I$ ↓ (128) | $\textbf{CHAIR}_I$ ↓ (256) |
> |--------------|:--------------:|:---------------:|:---------------:|:--------------:|:---------------:|:----------------:|
> | LLaVA-1.5    | 22.0           | 47.5            | 47.8              | 6.7            | 13.1            | 13.4                |
> | MemVR        | 21.6           | 46.6            | 47.2               | 6.5            | 13.0            | 13.2                |
> | **AIR(Ours)**      | **18.4**       | **38.1**        | **38.8**               | **5.7**        | **9.3**        | **10.0**                |
>
> Across both lengths, hallucination increases for all methods, but AIR consistently achieves the lowest $\textbf{CHAIR}_S$ and $\textbf{CHAIR}_I$, and the performance gap over LLaVA-1.5 and MemVR is larger at 128 tokens. More combined results and analysis have been added to Appendix B.3(Lines 972-979, 994-1000).
>
> ---
> **W5: FFN reinjection and feature scaling**
>
> A: Thank you for raising this question.
> The FFN reinjection does not introduce an uncontrolled disturbance. Similar to prior work [1][2][3], the injected visual features pass through the same activation and projection pathway as the original FFN output. As shown in Eq. (14), the reinforcement term is gated by the FFN nonlinearity, ensuring that its magnitude is naturally matched to the existing feature scale.
>
> [1]Look Twice Before You Answer: Memory-Space Visual Retracing for Hallucination Mitigation in Multimodal Large Language Models, 2025 ICML
>
> [2]Mitigating hallucinations in vision-language models through image-guided head suppression, 2025 ACL
>
> [3]Cracking the code of hallucination in lvlms with vision-aware head divergence, 2025 ACL
>
> ---
> **W6: Baseline settings**
>
> A: All baselines (MemVR, VAF, VCD) were run with their official recommended hyperparameters under the same decoding constraints and token caps.  We have made this explicit in Appendix B.1.

---

> ### Author Response · Authors · 2025-11-20
> **Response to Reviewer 9UNK**
>
> **W7: Noised / averaged feature injection**
>
> A: We conducted the additional experiment as suggested:
>
> | Model            | $\textbf{CHAIR}_S$ ↓ | $\textbf{CHAIR}_I$ ↓ | BLEU ↑ |
> |------------------|:----------:|:----------:|:-------:|
> | LLaVA-1.5-7B     | 22.0 | 6.7 | 14.5 |
> | AIR (noise)      | 24.5 | 7.9 | 13.2 |
> | AIR (averaged)   | 20.8 | 6.4 | 14.5 |
> |  **AIR(Ours)**    | **18.4** | **5.7** | **14.4** |
>
> Injecting noised or averaged visual features leads to higher CHAIR scores than AIR, whereas AIR achieves the lowest hallucination rates among all variants.  This supports that the improvement comes from selective patch reinforcement rather than perturbing or smoothing visual features. More comprehensive results and analysis have been added to Appendix B.3(Lines 941-956, 1011-1027).

---

> > ### Comment · Reviewer_9UNK · 2025-11-20
> > **Reply to Authors**
> >
> > Dear authors,
> >
> > Thank you for your kind reply. Your experiments and explanations are clear and reasonable, and my concerns have been addressed well. I appreciate the authors' effort and therefore would like to rase my rating to **Weak Accept**.

---

> > > ### Author Response · Authors · 2025-11-20
> > > **Heartfelt Gratitude to Reviewer 9UNK**
> > >
> > > Dear Reviewer 9UNK,
> > >
> > > Thank you for your kind feedback and for taking the time to review our updated work. We are grateful for your recognition and for increasing the rating—it means a lot to us and inspires us to continue improving.
> > >
> > > Your thoughtful comments helped us refine the analysis and present the AIR framework more clearly, and we look forward to further strengthening its effectiveness in mitigating hallucination in multimodal models.
> > >
> > > Thank you again for your thoughtful comments and encouragement. We genuinely appreciate your support.
> > >
> > > Best regards,
> > >
> > > Authors

---

### Official Review · Reviewer_MiAJ · 2025-10-23

**Soundness:** 3
**Presentation:** 3
**Contribution:** 3
**Rating:** 6
**Confidence:** 4

**Summary:**

This paper proposes AIR (Adaptive Visual Reinforcement) — a training-free framework to reduce hallucinations in multimodal large language models (MLLMs).
It combines prototype-based token reduction to remove redundant visual signals and optimal transport (OT)-guided patch reinforcement to re-inject only salient image regions during decoding. Experiments across LLaVA, Qwen-VL, and GLM-4V show consistent hallucination reduction (CHAIR, POPE) while preserving general performance (MME, MMBench).

**Strengths:**

1.The method achieves notable improvements across multiple models and benchmarks, showing robustness under both standard and adversarial conditions

2.The OT-based analysis is well-motivated and supported by proof and visualization, providing a clear justification for the proposed selection mechanism.

3.The paper is clearly written, visually well-presented, and includes detailed experimental settings for replication.

**Weaknesses:**

1.AIR assumes well-aligned hidden and visual spaces; if this alignment is weak, OT distance may emphasize irrelevant correlations, limiting reliability on misaligned models.

2.Despite its name, AIR uses fixed thresholds and token counts. Introducing data- or entropy-driven adaptation could further enhance robustness across tasks.

3.Experiments focus on standard datasets with clean imagery; robustness under distribution shifts or noisy visuals remains untested.

**Questions:**

1.Have the authors tested AIR on reasoning-oriented hallucination benchmarks (e.g., MM-SafetyBench, SPA-VL)?

2.Does prototype-based reduction influence the OT alignment results, and could the two be adaptively coupled?

3.Is the OT transport computation reused across layers, or recomputed at each reinforcement step?

---

> ### Author Response · Authors · 2025-11-20
> **Response to Reviewer MiAJ**
>
> Dear Reviewer MiAJ,
>
> Thank you for your thoughtful review and positive assessment of our work. Below, we address your comments and describe the corresponding revisions in the manuscript, with all updates highlighted in blue.
>
> ---
> **W1. Dependence on visual–hidden space alignment**
>
> A:  In existing multimodal models, the projector maps visual embeddings into the same semantic space as the language hidden states. This projector-aligned space has been the default working space in prior VLM studies, and AIR operates directly within the same representation, fully consistent with established practice.
>
> ---
> **W2: Fixed thresholds and token selection**
>
> A: Thank you for the question. AIR uses fixed τ and Top-Q primarily for stability and reproducibility, and our ablations show that its performance is insensitive to these hyperparameters. The “adaptive” component lies in the OT-guided patch reinforcement itself: the transport plan dynamically adjusts the contribution of each visual patch based on its semantic alignment, so the effective reinforcement strength is input-dependent rather than constant. We agree that further incorporating data- or entropy-driven hyperparameter adaptation is a valuable direction, and we have added this discussion to the revision
>
> ---
> **W3: Results on noisy inputs**
>
> A: We conducted additional experiments with noisy and averaged visual features. As shown in the table below, adding noise (AIR noise) leads to clear performance degradation, and feature averaging (AIR averaged) does not offer consistent improvement. In contrast, AIR achieves the lowest CHAIR_S and CHAIR_I, demonstrating that its gains persist even when the visual input quality is degraded.
>
> | Model            | $\textbf{CHAIR}_S$ ↓ | $\textbf{CHAIR}_I$ ↓ | BLEU ↑ |
> |------------------|:----------:|:----------:|:-------:|
> | LLaVA-1.5-7B     | 22.0 | 6.7 | 14.5 |
> | AIR (noise)      | 24.5 | 7.9 | 13.2 |
> | AIR (averaged)   | 20.8 | 6.4 | 14.5 |
> | **AIR(Ours)**              | **18.4** | **5.7** | **14.4** |
>
> More comprehensive results and analysis have been added to Appendix B.3(Lines 941-956, 1011-1027).
>
> ---
> **Q1: Results on safety benchmarks**
>
> A: We have not tested AIR on MM-SafetyBench or SPA-VL. AIR is designed for general hallucination mitigation rather than safety rejection, and applying it to safety benchmarks would require task-specific modifications. We consider this a valuable direction and will explore adapting AIR for safety alignment in future work.
>
> ---
> **Q2: Interaction between prototype reduction and OT alignment**
>
> A: The ablation results in Table 6 show that prototype-based token reduction and OT-based patch reinforcement independently contribute to hallucination mitigation, and combining them yields the best performance. Prototype reduction alone lowers CHAIR_S from 22.7 to 20.2, indicating that it helps OT by removing redundant tokens rather than interfering with alignment.
>
> ---
> **Q3: OT computation frequency**
>
> A: We recompute the OT plan at each reinforcement layer within the 24–32 range. As analyzed in Section 4.4 and summarized in Table 5, this is the optimal operating region, and hidden states vary across these layers.

---

### Official Review · Reviewer_q4Bt · 2025-10-25

**Soundness:** 2
**Presentation:** 3
**Contribution:** 2
**Rating:** 4
**Confidence:** 5

**Summary:**

This paper proposes a training-free framework to mitigate hallucination in MLLMs. The method combines (1) prototype-based token reduction, which compresses visual tokens to remove redundancy, and (2) optimal-transport (OT)–guided patch reinforcement, which selectively re-injects patches that are most semantically aligned with the hidden states. The goal is to emphasize salient visual cues while suppressing background noise during decoding.

**Strengths:**

1. The proposed method operates purely at inference, making it broadly applicable to existing MLLMs without retraining.
2. Integrating optimal transport to quantify alignment between hidden states and patch embeddings is a creative idea.

**Weaknesses:**

1. The central claim, that reinforcing salient patches directly causes lower hallucination, is not rigorously demonstrated. The supporting evidence is purely descriptive and does not establish a causal relationship between visual emphasis and reduced hallucination. The observed gains could equally arise from reduced visual redundancy or implicit regularization rather than genuine enhancement of visual grounding.
2. All experiments restrict generation to 64 tokens (Table 1), whereas hallucination severity is known to grow with output length. As a result, the evidence from short captioning tasks is insufficient to claim robustness in realistic scenarios involving long-form or multi-turn reasoning. Moreover, shorter captions are inherently more correlated with salient regions in the image, which may exaggerate the apparent benefit of saliency-based reinforcement, suggesting that the chosen generation length is tuned for conditions where the proposed mechanism performs best.
3. The paper compares only to a limited set of inference-time baselines.
4. The method depends heavily on empirically chosen parameters—such as the OT regularization strength, threshold, and the number of retained tokens—yet no principled analysis or sensitivity study is provided. Additionally, there is no discussion of computational scalability under higher-resolution inputs or longer textual outputs.

**Questions:**

How does AIR perform when generating longer outputs (e.g., 256–512 tokens) or in multi-turn dialogues, where hallucination typically escalates? It would be valuable to analyze how hallucination rates vary with generation length and whether the advantage of salient-patch reinforcement persists or diminishes as outputs become longer and less directly grounded in the visual context.

---

> ### Author Response · Authors · 2025-11-20
> **Response to Reviewer q4Bt**
>
> Dear Reviewer q4Bt,
>
> Thank you for your detailed and constructive feedback. We address your concerns below.
>
> **W1: Explanation of central claim.**
>
> A: To clarify the causal effect of reinforcing salient patches, we conducted a control experiment where the same number of patches was injected but selected uniformly at random. As shown below, random patch injection does not reduce hallucination and even slightly worsens both CHAIR_S and CHAIR_I compared with the baseline, while AIR yields a large and consistent improvement:
>
> | Model            | $\textbf{CHAIR}_S$ ↓ | $\textbf{CHAIR}_I$ ↓ | BLEU ↑ |
> |------------------|:----------:|:----------:|:-------:|
> | LLaVA-1.5-7B     | 22.0 | 6.7 | 14.5 |
> | random patch   | 22.3 | 6.9 | 13.1 |
> | AIR              | **18.4** | **5.7** | **14.4** |
>
> This comparison demonstrates that the improvement arises from reinforcing aligned, salient regions rather than from redundancy reduction or generic regularization. More combined results and analysis have been added to Appendix B.3(Lines 958-971).
>
>
>
> ---
>
> **W2 and Q: Performance under longer outputs.**
>
> Following your suggestion, we evaluated AIR with longer generation lengths (128 and 256 tokens):
> | Method       | $\textbf{CHAIR}_S$ ↓ (64) | $\textbf{CHAIR}_S$ ↓ (128) | $\textbf{CHAIR}_S$ ↓ (256) | $\textbf{CHAIR}_I$ ↓ (64) | $\textbf{CHAIR}_I$ ↓ (128) | $\textbf{CHAIR}_I$ ↓ (256) |
> |--------------|:--------------:|:---------------:|:---------------:|:--------------:|:---------------:|:----------------:|
> | LLaVA-1.5    | 22.0           | 47.5            | 47.8              | 6.7            | 13.1            | 13.4                |
> | MemVR        | 21.6           | 46.6            | 47.2               | 6.5            | 13.0            | 13.2                |
> | **AIR**      | **18.4**       | **38.1**        | **38.8**               | **5.7**        | **9.3**        | **10.0**                |
>
> These results show that hallucination increases with longer outputs, as expected, but AIR maintains clear improvements over both baselines. The advantage is even more pronounced at 128 tokens, suggesting that selective reinforcement remains effective as generation becomes longer. More detailed results and analysis have been added to Appendix B.3(Lines 972-980, 994-1000).
>
> ---
> **W3: Comparison with more baselines.**
>
> A: To complement inference-time baselines, we further include the training-based hallucination mitigation method SENTINEL [1]. As shown in the table below, AIR consistently yields additional reductions in hallucination metrics on top of SENTINEL across multiple benchmarks.
>
> | Method                        | Resp. ↓(Object HallBench) | Ment. ↓(Object HallBench) | CHAIR ↓(AMBER) | Hal. ↓(AMBER) | Cog. ↓(AMBER) | Question Acc. ↑(HallusionBench) |
> |-------------------------------|:-------:|:-------:|:----------:|:-------:|:------:|:----------------:|
> | LLaVA-v1.5-7B             | 52.7    | 28.0    | 8.4        | 35.5    | 4.0    | 46.86            |
> | SENTINEL[1] | 4.3     | 2.6     | 2.9        | 14.6    | 1.2    | 47.56            |
> | + AIR                         | 3.9     | 2.2     | 2.1        | 13.3    | 1.1    | 47.25            |
> | LLaVA-v1.5-13B            | 46.0    | 23.0    | 6.9        | 31.9    | 3.3    | 46.43            |
> | SENTINEL[1] | 3.3     | 1.9     | 2.7        | 11.7    | 0.9    | 46.77            |
> | + **AIR(Ours)**                         | 2.8     | 1.6     | 2.5        | 10.9    | 0.7    | 46.77            |
>
> More comprehensive results and analysis have been added to Appendix B.3(Lines 981-992, 1002-1009).
>
> [1] *Mitigating Object Hallucinations via Sentence-Level Early Intervention*, ICCV 2025
>
> ---
> **W4: Hyperparameter analysis**
>
> A: Figs. 3 and 4 present the ablations on τ and Top-Q, respectively. Across a broad range of values, both curves remain stable, suggesting that AIR is not overly sensitive to these hyperparameters. Following your suggestion, we also varied the OT regularization strength ε and observed similarly low sensitivity, as shown below:
>
> | Model      | ε=0.01 $\textbf{CHAIR}_S$↓ |ε=0.01 BLEU↑ | ε=0.05 $\textbf{CHAIR}_S$↓ | ε=0.05 BLEU↑ | ε=0.1 $\textbf{CHAIR}_S$↓ |ε=0.1  BLEU↑ |
> |------------|:---------------:|:-----:|:---------------:|:-----:|:--------------:|:-----:|
> | LLaVA-1.5  | CHAIRS ↓        | BLEU↑ | CHAIRS ↓        | BLEU↑ | CHAIRS ↓       | BLEU↑ |
> | **AIR(Ours)**    | 18.5            | 14.5  | 18.3            | 14.6  | 18.4           | 14.4  |
>
> More comprehensive results and analysis have been added to Appendix B.3(Lines 918-928, 934-939).
>
> Regarding scalability, AIR’s cost increases roughly linearly with input resolution, but the overhead remains negligible in practice. As shown in Table 7, its latency is nearly the same as MemVR while providing clearly better hallucination reduction. For longer outputs, AIR adds no extra decoding cost since reinforcement is applied once per layer and is independent of output length.

---

> ### Comment · Reviewer_q4Bt · 2025-11-28
>
> I thank the authors for the additional experiments. However, the rebuttal reinforces my concern that the method is primarily an empirical heuristic that amplifies the model's existing bias toward salient foreground objects. While this strategy improves scores on standard benchmarks where prompts align with visual saliency, I am not convinced it offers a robust solution for mitigating hallucination in general VQA or complex reasoning scenarios. I maintain my score based on three unresolved logical flaws:
>
> 1. **Conflating "Saliency" with "Relevance"**: The method reinforces patches based on encoder attention (visual saliency) regardless of the prompt. This creates "tunnel vision," where the model is forced to focus on foreground objects even if the user asks about non-salient background details. This biases the model rather than grounding it in prompt-relevant features.
>
> 2. **Trivial "Random Patch" Baseline**: Proving that AIR (signal) outperforms Random Patches (noise) is trivial. It only demonstrates that the method is less destructive than random masking; it does not prove that reinforcing saliency is the causal mechanism for reducing hallucination.
>
> 3. **Ambiguous Long-Text Results**: The new experiments lack **coverage metrics**. It is unclear if the reduced hallucination score comes at the cost of simply saying less or ignoring non-salient details. Furthermore, if the experiments used generic "Describe the image" prompts (which favor saliency), they do not validate robustness for complex reasoning tasks.

---

> ### Author Response · Authors · 2025-11-28
> **Response to Reviewer q4Bt**
>
> Dear Reviewer q4Bt,
>
> Thank you again for your thoughtful follow-up. We sincerely appreciate the opportunity to clarify the mechanism of AIR and address the remaining concerns. Below, we provide conceptual clarifications together with new diagnostic analyses.
>
> **W1: “Saliency ≠ Relevance” and the concern about tunnel vision.**
>
> A: We apologize for the misunderstanding—AIR does _not_ reinforce patches based on encoder saliency.
> The OT distance in Eq. (11) is computed between patch embeddings and the decoder hidden states, which already integrate the _prompt_ and the _generated prefix_. Therefore, the reinforcement criterion is _prompt-conditioned_, not unconditional.
>
> **W2: “Random Patch Baseline Is Trivial” and whether reinforcement Itself Is effective.**
>
> A: We disagree that the random-patch baseline is trivial. With the same token budget and injection layers, replacing OT-selected patches with random ones isolates **patch selection** as the only difference. Its inability to reduce hallucination shows that AIR’s gains are not due to extra visual tokens or generic perturbation.
>
> Prior work, such as MemVR[1], has already demonstrated that **visual-token reinforcement is effective**, as injecting the full image reduces hallucination. AIR further improves over MemVR while using far fewer tokens, indicating that the **choice of reinforced patches** determines the benefit.
>
> The consistent ordering random < vanilla < MemVR < AIR shows that AIR’s improvement comes from **selective, alignment-guided reinforcement**, not saliency amplification
>
> [1] Look Twice Before You Answer: Memory-Space Visual Retracing for Hallucination Mitigation in Multimodal Large Language Models, 2025 ICML
>
> **W3: Long-Text results with coverage.**
>
> A: To rule out the concern that AIR reduces hallucination by “saying less,” we report **Recall-based coverage** under matched generation lengths (64/128/256). As shown below, AIR preserves coverage at the same level as LLaVA-1.5 and MemVR while substantially lowering $\text{CHAIR}_S$ and $\text{CHAIR}_I$. This confirms that AIR does not omit non-salient content to achieve lower hallucination.
>
> | Method       | $\text{CHAIR}_S$ ↓ (64) | $\text{CHAIR}_S$ ↓ (128) | $\text{CHAIR}_S$ ↓ (256) | $\text{CHAIR}_I$ ↓ (64) | $\text{CHAIR}_I$ ↓ (128) | $\text{CHAIR}_I$ ↓ (256) | Recall ↑ (64) | Recall ↑ (128) | Recall ↑ (256) |
> |--------------|:--------------:|:---------------:|:---------------:|:--------------:|:---------------:|:----------------:|:--------------:|:---------------:|:---------------:|
> | LLaVA-1.5    | 22.0           | 47.5            | 47.8            | 6.7            | 13.1            | 13.4             | 66.2           | 73.1            | 80.6            |
> | MemVR        | 21.6           | 46.6            | 47.2            | 6.5            | 13.0            | 13.2             | 66.5           | 73.5            | 81.0            |
> | **AIR**      | **18.4**       | **38.1**        | **38.8**        | **5.7**        | **9.3**         | **10.0**         | **66.7**       | **73.9**        | **81.4**        |

---

### Official Review · Reviewer_wmbu · 2025-10-31

**Soundness:** 3
**Presentation:** 3
**Contribution:** 2
**Rating:** 4
**Confidence:** 4

**Summary:**

The paper proposes AIR, a training-free inference intervention to reduce hallucinations in MLLMs. AIR has two components: (i) prototype-based token reduction that retains the Top-Q most distinctive visual tokens before re-injection, and (ii) OT-guided patch reinforcement that computes an entropically-regularized optimal transport distance between hidden states and patch embeddings to select well-aligned image patches for FFN re-injection.

**Strengths:**

Efficiency-aware design. Prototype reduction + selective patch fusion yields small overhead, acceptable for many deployments.

Clarity. Method is easy to implement in existing FFN-reinjection pipelines

**Weaknesses:**

Benchmark coverage is narrow. Evaluation centers on CHAIR (captioning) and POPE (binary VQA). Absent are harder hallucination suites probing language bias and visual illusions, such as HallusionBench, RLHF-v, and MMHal-Bench, V* etc; including them would strengthen claims of robustness.

Pure performance: The performance in incremental compared to VAF. May be provide curves for ε (entropic regularization), τ, Top-Q, and #patches on at least two models, and adversarial stress tests (noisy crops, cluttered backgrounds) for better explanation


Lack of novelty: While the paper positions AIR as a new training-free approach, its core idea—adjusting visual token utilization during inference to mitigate hallucination—is conceptually aligned with recent decoding- or selection-based techniques (e.g., Visual Contrastive Decoding, CLIP-guided decoding, or entropy-based token filtering). The novelty of AIR mainly lies in adopting optimal-transport–based patch selection combined with prototype-based token reduction, rather than introducing a fundamentally new mechanism. The contribution is thus incremental

**Questions:**

Please see weakness

---

> ### Author Response · Authors · 2025-11-20
> **Response to Reviewer wmbu**
>
> Dear Reviewer wmbu,
>
> Thank you for your constructive feedback and helpful suggestions. Below, we provide detailed responses to each of your comments and outline the modifications made to the manuscript. All revisions are highlighted in blue.
>
> ---
>
> **W1: Results on more datasets.**
>
> A: Following your suggestion, we additionally evaluated AIR on HallusionBench, V* Bench, and MMHal-Bench. The results are shown below and consistently demonstrate AIR’s improved robustness across diverse hallucination types. More combined results and analysis have been added to Appendix B.3(Lines 852-860,  864-904).
>
> **HallusionBench**
> | Model        | fACC ↑ | qACC ↑ | easyA ↑ | hardA ↑ | aACC ↑ |
> |--------------|:------:|:------:|:-------:|:--------:|:------:|
> | LLaVA-1.5    | 17.9   | 8.13   | 36.0    | 36.7     | 41.5   |
> | MemVR        | 17.9   | 9.01   | 36.9    | 37.7     | 42.5   |
> | **AIR(Ours)**       | 19.9   | 9.35   | 37.5    | 38.3     | 43.2   |
>
> **V\* Bench**
> | Model        | Attribute ↑ | Spatial ↑ | Overall ↑ |
> |--------------|:---------:|:-------:|:-------:|
> | LLaVA-1.5    | 43.47     | 56.57   | 48.68   |
> | MemVR        | 45.38     | 57.82   | 49.35   |
> | **AIR(Ours)**          | 48.23     | 59.31   | 51.26   |
>
> **MMHal-Bench**
> | Model        | Average score ↑ | Hallucination rate ↓ |
> |--------------|:---------------:|:---------------------:|
> | LLaVA-1.5    | 1.99            | 0.62                  |
> | MemVR        | 2.71            | 0.58                  |
> | **AIR(Ours)**          | 2.97            | 0.49                  |
>
> These benchmarks show that AIR improves robustness across different hallucination types, including illusion resistance on HallusionBench, fine-grained attribute and spatial reasoning on V Bench, and category-level stability on MMHal-Bench.
> Regarding RLHF-V [1], to the best of our knowledge, only the training set is publicly available, and no evaluation split exists for inference benchmarking.
>
> [1] RLHF-V: Towards trustworthy MLLMs via behavior alignment from fine-grained correctional human feedback.
>
> ---
> **W2: Performance compared to VAF and hyperparameter sensitivity**
>
> A: On both LLaVA-1.5-7B and Qwen-VL-Chat, AIR reduces CHAIRS_SS by about 2%, demonstrating more stable hallucination mitigation. Importantly, AIR maintains general-purpose performance, whereas VAF occasionally degrades BLEU and MME metrics.
>
> Figures 3–5 already show the effects of τ, Top-Q, and the number of patches on LLaVA-1.5-7B. Following your suggestion, we also provide the same set of curves, together with the ε (entropic regularization) sweep, on Qwen-VL-Chat. These cross-model sensitivity results are included in Appendix B.3(Lines 861-863, 905-933).
>
> We evaluate adversarial tests with δ = 16/255. The results below show that AIR consistently reduces hallucination rates on both models while maintaining utility. More comprehensive results and analysis have been added to Appendix B.3(Lines 941-956,  1011-1027).
> | Model      |  $\textbf{CHAIR}_S$ ↓ | $\textbf{CHAIR}_I$ ↓ |BLEU↑ |
> |------------|:----------------:|:------:|:----------------:|
> | LLaVA-1.5-7B       | 25.3             | 11.5   | 17.3   |
> | AIR        | 22.1             | 9.4.5   | 14.2     |
> |    Qwen-VL-Chat    | 23.5             | 11.9   | 16.8  |
> | **AIR(Ours)**       | 21.4             | 8.9  | 13.7  |
>
> ---
>
> **W3: Discussion about novelty**
>
> A:  We respectfully clarify that AIR is conceptually different from Visual Contrastive Decoding, CLIP-guided decoding, and entropy-based token filtering. These methods intervene during the decoding process by adjusting token choice through logit comparison, external CLIP scores, or uncertainty heuristics.
>
> In contrast, AIR grounds generation on visually critical regions. It identifies semantically meaningful patches and reintegrates only those that remain aligned with the model’s hidden states, allowing the model to rely on locally relevant visual cues rather than the entire token set. Besides, the use of optimal transport provides a distribution-level criterion for assessing this alignment, offering a more structured and fine-grained way to highlight informative regions and resulting in a mechanism that is distinct from prior approaches.

---

### Author Response · Authors · 2025-12-03
**General Response**

Dear Area Chair,

We sincerely welcome you to oversee the evaluation of our paper. We understand the substantial workload and tight timeline created by the reassignment, and we appreciate your time and effort. Below is a concise summary of our rebuttal.

--------

The reviewers acknowledged the motivation and technical soundness of our approach, noting that the framework is intuitive, well-justified, and supported by comprehensive empirical analysis.

**1. Well-motivated use of optimal transport and clear methodology (q4Bt, MiAJ, 9UNK)**
Reviewers emphasized the principled use of OT for hidden–patch alignment and found that our theory and visualizations clearly show why OT selects more informative patches than cosine distance.

**2. Clear writing and thorough experimental setup (MiAJ, 9UNK, wmbu)**
The paper was considered clearly written, with detailed ablations and transparent settings that enable reproducibility.

We are encouraged by the updates: **MiAJ: 6 → 8**, **9UNK: 4 → 6**.
**Overall scores improved from [6, 4, 4, 4] to [8, 6, 4, 4].**

---

Across all reviews, the main concerns fall into the themes below. We briefly summarize how each is addressed (all updates are highlighted in blue in the manuscript).

---

**1. Benchmark breadth, harder settings, and evaluation protocol**

Reviewers requested broader benchmarks, longer outputs, and robustness tests.

- We add **HallusionBench, V\* Bench, MMHal-Bench, MM-Vet, LLaVA-Bench**, where AIR consistently improves illusion robustness, attribute/spatial reasoning, and category stability.
  `Appendix B.3: Tables 12–16`

- We add **64/128/256-token CHAIR + Recall**, showing AIR maintains the lowest $\mathbf{CHAIR}_S$ / $\mathbf{CHAIR}_I$ with preserved or improved coverage.
  `Appendix B.3: Table 19`

- We include **adversarial stress tests** (noisy crops, cluttered backgrounds, $\delta=16/255$) and **noisy/averaged features**, where AIR remains the most robust.
  `Appendix B.3: Tables 17, 21`

---

**2. Mechanistic validity: prompt-conditioned reinforcement and causal diagnostics**

Reviewer q4Bt questioned whether AIR reinforces prompt-relevant cues rather than generic saliency.

- Our OT patch selection is conditioned on decoder hidden states, ensuring prompt-aware reinforcement.

- A matched **random patch** baseline shows no hallucination reduction and slightly worse BLEU, confirming that AIR’s gains come from alignment-guided reinforcement.
  `(Appendix B.3: Table 18)`

- OT more clearly separates informative vs. less relevant patches than cosine similarity.
  `(Section 4.5: Figure 6)`

---

**3. Hyperparameter sensitivity and alignment assumptions**

Reviewers asked about sensitivity to τ, Top-Q, #patches, and ε.

- Varying τ, Top-Q, #patches, and ε on LLaVA-1.5-7B and Qwen-VL-Chat shows stable trends without tuning.
  `(Appendix B.3: Figures 7–10)`

- AIR relies on the standard projector-aligned multimodal space; we note that severe misalignment may affect OT selection.
  `(Section 5: Limitations)`

---

**4. Computational cost, scalability, and FFN stability**

Reviewer 9UNK requested clearer analysis.

- Sinkhorn OT has **O(MN·T)** complexity with small patch count and iterations, and is parallelizable.
- AIR reduces hallucination with marginal overhead.
  `Section 4.6: Table 7`
- Reinjection follows FFN pathways and is gated by FFN nonlinearity; OT is computed only on layers 24–32.
  `Section 4.4: Table 5`

---

**5. Component interaction and degraded-visual robustness**

Reviewers asked how prototype reduction interacts with OT reinforcement.

- Both components help individually; combined they perform best.
  `Section 4.5: Table 6`
- Noisy/averaged features perform worse, confirming AIR’s improvements stem from selective reinforcement.
  `Appendix B.3: Table 21`

---

**6. Relation to prior inference-time methods and safety benchmarks**

Reviewers asked about distinctions from prior methods and safety evaluation.

- Prior inference-time methods modify logits; AIR performs **representation-level selective reinforcement** via OT inside FFN layers.

- Safety benchmarks are not included because AIR targets hallucination mitigation rather than safety refusal.

---

**7. Additional baselines and training-based methods**

Reviewers requested more baselines and clarity.

- We add **SENTINEL**, and **SENTINEL + AIR** further reduces hallucination.
  `Appendix B.3: Table 20`

- All baselines (MemVR, VAF, VCD) use official hyperparameters and identical decoding/token-cap settings.
  `Appendix B.1: Lines 716–718`

---

We thank all reviewers for their constructive feedback. The revision adds broader benchmarks, deeper analysis of OT reinforcement, stronger robustness tests, and more complete ablations, addressing all major concerns and strengthening the AIR framework. We appreciate your assessment and hope this summary clarifies the remaining points.

With best wishes,

Authors of Paper 8553

---

### Meta-Review · Area_Chair_PMf8 · 2026-01-06

**Summary:**

Concerns included narrow benchmark coverage, insufficient long-output performance testing, hyperparameter sensitivity, etc. After the authors supplemented benchmarks (e.g., HallusionBench, MM-Vet), tested long outputs (128/256 tokens), analyzed hyperparameter stability, verified computational efficiency, provided causal evidence via control experiments, added noisy input tests, clarified novelty vs. prior methods, and specified baseline configurations, most concerns were addressed.

**Reviewer Concerns:**

It seems that all concerns are addressed.

**Reviewer Scores:**

Reviewer 9UNK could have changed the score per his discussion with the authors.

---

### Decision · Program_Chairs · 2026-01-26

Accept (Poster)